# The Sign Estimator: LLM Alignment in the Face of Choice Heterogeneity

## Abstract

Traditional LLM alignment methods are vulnerable to heterogeneity in human preferences. Fitting a naïve probabilistic model to pairwise comparison data (say over prompt-completion pairs) yields an inconsistent estimate of the population-average utility—a canonical measure of social welfare. We propose a new method, dubbed the *sign estimator*, that provides a simple, provably consistent, and efficient estimator by replacing cross-entropy with binary classification loss in the aggregation step. This simple modification recovers consistent ordinal alignment under mild assumptions and achieves the first polynomial finite-sample error bounds in this setting. Using standard benchmark experiments and a new empirical methodology to assess the impact of heterogeneity, we find that the sign estimator substantially reduces preference distortion compared to standard RLHF. Specifically, it cuts disagreement with true population preferences from 12% to 8%, and reduces angular estimation error by nearly 35%. Our empirical set-up leverages digital twins—LLMs calibrated to real-world US panelists—to simulate realistic population-level heterogeneity and obtain a ground truth alignment target for evaluating different estimators. Our estimator also compares favorably to panel data heuristics that explicitly model user heterogeneity and require tracking individual-level preference data—all while maintaining the implementation simplicity of existing LLM alignment pipelines.

## 1 Introduction

We consider the problem of learning a reward (or utility) model, say, in the context of LLM alignment. We are particularly concerned with doing so in the face of user heterogeneity.[1] Our problem can be described as follows: we are given a set of pairs of alternatives (for instance, these may be pairs of text completions for different prompts). For each pair, we collect feedback from a random user on which alternative they prefer. We seek to aggregate the user feedback by learning the *population-average utility* for each alternative.

In the typical generative model for the above setup, given a pair of alternatives $x, x'$ in a set $\mathcal{X}$, a random user has random utility $u(x)+\xi$ (respectively, $u(x')+\xi'$) for the two alternatives and selects the alternative with the higher random utility. Here $u(\cdot) : \mathcal{X} \to \mathbb{R}$ is a utility function that we seek to learn and $\xi, \xi'$ are i.i.d. noise terms. Given the data collected from user choices, the task of learning $u(\cdot)$ is relatively straightforward and essentially accomplished via maximum likelihood estimation.

The above model does not capture systematic heterogeneity in user preferences—a feature of the real world. A typical view of the setup with heterogeneity would associate a user with a parameter $\beta$ in a set $\mathcal{B}$ and model the user's random utility for $x$ via the random utility function $u(x; \beta) + \xi$. Given users drawn from a distribution over $\mathcal{B}$, a canonical learning

---

[1]Heterogeneity is ubiquitous in human feedback data used for LLM post-training. For example, on the Helpfulness-Harmlessness dataset with trained annotators, Bai et al. (2022b) reports only 63% agreement between trained annotators and Anthropic researchers.

task would be to recover $\bar{u}(\cdot) \triangleq \mathbb{E}[u(\cdot; \beta)]$. Aligning according to this utility function would be equivalent to aligning according to the average utility of the population.[2]

**The Problem:** What if we attempted to learn $\bar{u}(\cdot)$ while being oblivious to the underlying heterogeneity in the user population? That is, what if we assumed individual utility models took the simplified form $u(\cdot)$ rather than $u(\cdot; \beta)$? In effect, this is what existing reward learning pipelines do. As it turns out, the results of doing this yield a substantively biased view of population preferences and fail to recover $\bar{u}(\cdot)$. In fact, these biases can result in perverse impacts on model behavior, as we demonstrate later. More troubling still, recovering $\bar{u}(\cdot)$ is, in general, impossible (Gölz et al., 2025). Our problem, therefore, is identifying a meaningful set of utility models and a learning algorithm that recovers $\bar{u}(\cdot)$ or a close approximation thereof.

## 1.1 THIS PAPER

As stated above, our focus is on the task of learning $\bar{u}$ in the face of user heterogeneity. In the important special case of *linearly parametrized* utility models where $u(x; \beta) = \langle \phi(x), \beta \rangle$, which is typical in RLHF as it amounts to appending a linear head $\beta$ to the hidden representation of a pretrained language model $\phi(x)$. This reduces to the task of recovering $\bar{\beta} = \mathbb{E}_\beta[\beta]$. We make the following contributions:

**Mis-specified Estimation:** We show that the status-quo approach to our learning problem in Reinforcement Learning from Human Feedback(RLHF) pipelines (essentially, MLE on a mis-specified model) can be interpreted as a certain "re-weighting" of $\beta$ that under-weights low variance preferences; this transparently characterizes the nature of the bias introduced by the typical approach to learning $\bar{u}$.

**The Sign Estimator:** We introduce the *Sign estimator*. We show that when user heterogeneity is symmetric in a sense we make precise later, the population level sign estimator recovers identical ordinal preferences to $\bar{u}$. In the context of linear utility models, we recover $\bar{\beta}/\|\bar{\beta}\|$. Importantly, our assumptions subsume the typical assumptions made regarding user preferences in common econometric models of heterogeneous preferences.

**Polynomial Rates:** In the context of linear utility models, the sign estimator recovers $\bar{\beta}/\|\bar{\beta}\|$ at a $O(n^{-1/3})$ rate. This is a significant step forward: the best estimators for this problem in the generality we allow have error rates where the exponent typically deteriorates with the dimension $d$.

**Practical Use and Performance:** The sign estimator is practical to deploy; it serves as a *drop-in replacement* for cross-entropy loss in reward learning pipelines. On real-world human feedback datasets and simulated preference data collected from a set of digital twins calibrated to real users and without assuming symmetry, we show that the use of the Sign estimator reduces the error in accuracy by 40%, cuts disagreement rates from 12% to 8%, and reduces angular error from 63° to 41° when compared with the status-quo approach to reward learning.

## 1.2 RELATED LITERATURE

The success of LLM systems has spurred significant work on modeling and optimizing RLHF pipelines (Bai et al., 2022a; Ouyang et al., 2022a; Ziegler et al., 2020a). In the standard procedure, a reward model aggregates human preferences via maximum-likelihood estimation—under Bradley–Terry or Plackett–Luce assumptions—and then yields a fine-tuned policy (Ouyang et al., 2022b; Zhu et al., 2023; Wang et al., 2023). Reward models, however, fail to capture heterogeneous views in the population—a point widely recognized in recent literature (Xu et al.)—distorting the utilitarian objective. For example, Siththaranjan et al. (2023) proves that the standard reward learning approach implements a social-choice rule known as "Borda count", which is less intuitive and lacks strong axiomatic justification. Our work provides an intuitive analysis of this inconsistency, making explicit how this estimator implicitly re-weights individuals in its aggregation process.

---

[2]This notion of social welfare is justified via Harsanyi's Utilitarian theorem (Harsanyi, 1955); see Appendix B.

We focus on average-utility maximization over the population as a social choice rule. This coincides with the standard RLHF objective when preferences are homogeneous and is consistent with welfare analysis in economics.[3] The best-known results in this context are competitive algorithms—incurring a constant-factor "distortion" in the worst case (Gölz et al., 2025). In contrast, our sign estimator provably converges to the population-average ordinal preferences under a mild assumption of symmetry—a condition that holds for example in the case of the mixed logits model.

Recent literature proposes pluralistic alignment methods such as estimation of mixture models or even personalized alignment (Park et al., 2024a; Scheid et al., 2024; Chakraborty et al., 2024; Poddar et al., 2024), which are either computationally more involved or alter the standard RLHF architecture. Our work also contrasts with recent literature that proposes axiomatic reformulations of LLM alignment (Ge et al., 2024) or game-theoretic solution concepts (Munos et al., 2023) to mitigate the limitations of reward models.

Lastly, our estimation results contribute to inference for mixtures of generalized linear models—a notoriously difficult problem (Ammar et al., 2014; Oh & Shah, 2014; Sedghi et al., 2016) which often requires panel data structures (Kallus & Udell, 2020). Remarkably, our method circumvents the need to actually estimate the mixture, which is often unidentifiable, and instead directly recovers the population-average utility direction, which is identified under less restrictive assumptions (Fox et al., 2012).

## 2 PROBLEM SET-UP

As discussed in the Introduction, we are given a potential set of alternatives $\mathcal{X}$; as a concrete example, $\mathcal{X}$ could represent the set of all (prompt, completion) pairs. A user is parameterized by a parameter $\beta \in \mathcal{B}$. We are also endowed with a distribution over users (i.e. over $\mathcal{B}$) and when clear from context will also use $\beta$ to denote the parameter of a random user drawn from $\mathcal{B}$ according to this distribution.

**Utility Model:** A user with parameter $\beta$ is assumed to have a mean utility function parameterized by $\beta$, $u(\cdot; \beta) : \mathcal{X} \to \mathbb{R}$. Such a user ascribes random utility $u(x; \beta) + \xi_{x,\beta}$ to alternative $x$. We assume $\{\xi_{x,\beta} : x \in \mathcal{X}, \beta \in \mathcal{B}\}$ to be an i.i.d. collection of mean-centered Gumbel (i.e., Gumbel$(-\gamma, 1)$) random variables. We assume the existence of a distinct alternative $o \in \mathcal{X}$ with $u(o; \beta) = 0$ for all $\beta \in \mathcal{B}$; $o$ is often referred to as the 'outside option'. Given a pair of alternatives $x_1, x_2 \in \mathcal{X}$, a user $\beta$ thus selects the first alternative with probability $\sigma(u(x_1; \beta) - u(x_2; \beta))$ where $\sigma(\cdot)$ is the Logistic function. This is known as the Bradley-Terry or Plackett-Luce model (**?**Luce, 1959).

An important focal point of our work will be the case of *linear utility functions* $u(\cdot; \beta) = \langle \phi(\cdot), \beta \rangle$ where, $\phi(\cdot)$ is a general feature map (e.g., last hidden layer of a neural network) and $\beta$ is the user's utility vector. Of course, the linearity assumption is without loss if we allow for general feature maps, which are nonlinear in the input covariates. Our approach is fully agnostic to the downstream RLHF policy, meaning that it imposes no restriction on the language model that is post-trained.

**Data:** We assume a preference dataset constructed via the following generative process: We sample a random user $\beta$ from $\mathcal{B}$. We also sample a pair of alternatives $(X_1, X_2)$ for the same prompt according to a distribution $p$ over $\mathcal{X}^2$. We assume $p$ is exchangeable (so that the order of alternatives conveys no information) and to avoid certain degeneracies, we also assume that the set of ties $\{(x_1, x_2) \in \mathcal{X}^2 : \mathbb{E}_\beta[u(x_1; \beta)] = \mathbb{E}_\beta[u(x_2; \beta)]\}$ has measure zero. We observe $Y \triangleq \mathbf{1}\{u(X_1; \beta) + \xi_{X_1, \beta} \geq u(X_2; \beta) + \xi_{X_2, \beta}\}$, i.e., whether the user prefers $X_1$ over $X_2$. Our preference dataset is $\{(X_1, X_2)_i, Y_i : i \in [n]\}$, a collection of $n$ such i.i.d. draws. Notice that in this setup, conditioned on $(X_1, X_2)$, $Y$ is distributed as a Bernoulli random variable with mean $\mathbb{E}_\beta[\sigma(u(X_1; \beta) - u(X_2; \beta))]$.

**Learning Task:** Given a preference dataset, our goal is to recover the mean utility function $\bar{u}(\cdot) \triangleq \mathbb{E}_\beta[u(\cdot; \beta)]$, corresponding to the utilitarian aggregate of the population preferences.

---

[3]While welfare functionals can be more general, the additive utilitarian form has been predominant since Harsanyi's aggregation result (Harsanyi, 1955).

In the case of a linear utility function, we simply care to recover the mean utility vector $\bar{\beta} \triangleq \mathbb{E}_\beta[\beta]$.

In this paper we will satisfy ourselves with the goal of recovering $\bar{u}(\cdot)$ up to *induced preferences*: we will recover a $u : \mathcal{X} \mapsto \mathbb{R}$ for which $u(x_1) \geq u(x_2) \iff \bar{u}(x_1) \geq \bar{u}(x_2)$ for any $x_1, x_2 \in \mathcal{X}$. If $\mathcal{X}$ has a non-zero Lebesgue measure in $\mathbb{R}^d$, this amounts to recovering the direction of the average utility vector $\bar{\beta}/\|\bar{\beta}\|$. In particular, this learning target suffices to determine the class of fine-tuned policies via RLHF where one optimizes a linear combination of expected utility and KL divergence to a reference policy; see Appendix B. We clarify, however, that the estimation methods considered in this paper return a reward model; only our theoretical guarantee holds up to induced preferences.

## 2.1 Status Quo RLHF Estimator

The status-quo approach to the learning task above would simply seek to find a single utility function that minimizes cross-entropy loss to the observed preference data (Ziegler et al., 2020b; Ouyang et al., 2022a; Christiano et al., 2023). Specifically, given a (sufficiently rich) class of functions $\mathcal{U}$ such that $\bar{u} \in \mathcal{U}$, we estimate

$$\hat{\mu}^{\mathrm{RLHF}} \in \underset{\tilde{u} \in \mathcal{U}}{\arg\min} -\mathbb{E}\left[Y \log\left(\sigma(\tilde{u}(X_1) - \tilde{u}(X_2))\right) + (1 - Y) \log\left(1 - \sigma(\tilde{u}(X_1) - \tilde{u}(X_2))\right)\right].$$

By a slight abuse of terminology, we refer to the status-quo estimator as the *RLHF estimator* and note that it is predominantly employed in RLHF pipelines (Bai et al., 2022a; Askell et al., 2021; Ziegler et al., 2020a). We immediately observe that the loss function above is misspecified: specifically, conditioned on $(X_1, X_2)$, $Y$ is actually distributed as a Bernoulli with mean $\mathbb{E}_\beta[\sigma(u(X_1; \beta) - u(X_2; \beta))]$ whereas the above loss seeks to model $Y$ as a Bernoulli with mean $\sigma(\tilde{u}(X_1) - \tilde{u}(X_2))$. Can we nonetheless have $\hat{\mu}^{\mathrm{RLHF}} = \bar{u}$? We study this issue in the remainder of this section, but start with a simple example illustrating the inconsistency noticed in recent literature:

Consider two types of users, $\mathcal{B} = \{1, 2\}$, faced with two alternatives, $\mathcal{X} = \{o, 1\}$. Users of type 1 make up a fraction $\alpha$ of the population. We assume $u(1; 1) = 1$ while $u(1; 2) = -M < 0$. We can show that for any $M \geq 3$ and any $\alpha \in [0.7, 1 - 1/M]$, the RLHF estimator recovers $\tilde{u}(1) > 0$ while $\bar{u}(1) < 0$; i.e. the RLHF estimator recovers the wrong social preference; see Appendix C. Remarkably, if we keep $\alpha$ fixed (at say 0.7) and let $M$ grow large: despite 30% of the population having a strong preference for $o$ over option 1, with the remaining 70% having a relatively negligible preference for option 1, the RLHF estimator still picks option 1. In other words this is a setting where the RLHF estimator incurs arbitrarily large disutility to the population as a whole.

## 2.2 Utility Aggregation for Gaussian Data via the RLHF Estimator

Let us specialize here to linear utility models and denote $X \triangleq \phi(X_1) - \phi(X_2)$. The RLHF estimator learns $\hat{\beta}^{\mathrm{RLHF}}$ satisfying the moment equation

$$\mathbb{E}_X[\sigma(X^\top \hat{\beta}^{\mathrm{RLHF}})X] = \mathbb{E}_X[\mathbb{E}_\beta[\sigma(X^\top \beta)]X] .$$

This identity shows that the RLHF estimator calibrates $\hat{\beta}^{\mathrm{RLHF}}$ so that the choice frequencies $\sigma(X^\top \hat{\beta}^{\mathrm{RLHF}})$ match the observed choice frequencies $\mathbb{E}_\beta[\sigma(X^\top \beta)]$, weighted by the context $X$. Siththaranjan et al. (2023) show that this can be interpreted as an aggregation of individual preferences via the so-called Borda count. Here we take a complementary perspective that allows for a transparent interpretation of how the RLHF estimator aggregates utility functions (as opposed to preferences). We show that for Gaussian data, $\hat{\beta}^{\mathrm{RLHF}}$ can be interpreted as a *re-weighted* mean of $\beta$. This re-weighting introduces bias by overweighting certain types of users and under-weighting others:

**Proposition 1.** *Suppose that $(X_1, X_2)$ is distributed so that $\phi(X_1) - \phi(X_2) \triangleq X \sim \mathcal{N}(0, \Sigma)$. Then the RLHF estimator recovers $\hat{\beta}^{\mathrm{RLHF}}$ satisfying:*

$$\hat{\beta}^{\mathrm{RLHF}} \propto \mathbb{E}_\beta[\mathbb{E}_X[\sigma'(X^\top \beta)]\beta] \tag{1}$$

Proposition 1 reveals a fundamental issue with how the RLHF estimator aggregates heterogeneous utility vectors in the population. While our goal is to estimate the expected utility $\bar{\beta} = \mathbb{E}[\beta]$, the RLHF estimator instead recovers a weighted average $\mathbb{E}[w(\beta)\beta]$ where the (non-negative) weighting function $w(\beta) = \mathbb{E}_X[\sigma'(X^\top \beta)]$. This weighting scheme has a noteworthy interpretation: since $\sigma'(X^\top \beta) = \sigma(X^\top \beta)(1 - \sigma(X^\top \beta))$ is the variance of user $\beta$'s choice response for the pair $(X_1, X_2)$, the RLHF estimator amplifies the influence of uncertain users while diminishing that of confident ones. That is, if a user strongly favors one of the two outputs, their stated preferences are *discounted*! In fact this discounting is quite extreme: noting that $\lim_{\|\beta\| \to +\infty} \mathbb{E}_X[\sigma'(X^\top \beta)]\beta = 0$ we see that in the limit, the users who have negligible variance in their choices—or equivalently care most strongly (or have the most informed opinion) about a particular outcome—are ignored. This behavior seems undesirable. We note that this is *precisely* the behavior we saw in the simple example with two alternatives.

## 3 The Sign Estimator

The previous section illustrated that the RLHF estimator is inconsistent in general for recovering $\bar{u}$ (or $\bar{\beta}$ in the linear utility case). Further, it is known that it is impossible to recover $\bar{u}$ for general distributions on the random field $\epsilon(\cdot) \triangleq u(\cdot; \beta) - \bar{u}(\cdot)$; see Gölz et al. (2025). As such, this Section seeks to make progress via a mild assumption on $\epsilon(\cdot)$:

**Assumption 1.** *The distribution over individual utilities is symmetric about its mean: specifically, for all $x \in \mathcal{X}$, $\epsilon(x)$ and $-\epsilon(x)$ have the same distribution.*

We emphasize two points: first, the RLHF estimator remains biased even under the above assumption; see Appendix D.2. Second, and more importantly, the assumption of symmetry allows for the so-called Gaussian mixed logit family of random utility models. The latter family is the workhorse of modeling heterogeneous user preferences in the economics literature (Train, 2009, Chap. 6). The remainder of this Section develops an estimator we dub the *Sign estimator*, which we show to mitigate the problem of inconsistency. We begin with a simple observation that underpins our development of the sign estimator:

**Proposition 2.** *If $\epsilon(x)$ is symmetrically distributed for all $x \in \mathcal{X}$, we have*

$$\text{sign}\left(\bar{u}(x_1) - \bar{u}(x_2)\right) = \text{sign}\left(\mathbb{P}\left(Y = 1 | X_1 = x_1, X_2 = x_2\right) - \frac{1}{2}\right)$$

*Proof.* For $x_1, x_2 \in \mathcal{X}$ and $t \in \mathbb{R}$, define $g(t) \triangleq \mathbb{E}_\varepsilon[\sigma(t + \varepsilon(x_1) - \varepsilon(x_2))]$. The function $g$ is increasing since $\sigma$ is. Moreover, $g(0) = 1/2$. To see the latter note that $g(0) = \mathbb{E}_\varepsilon(\sigma(\varepsilon(x_1) - \varepsilon(x_2)))$. But by the assumed symmetry of $\epsilon(\cdot)$, and the identity $\sigma(t) = 1 - \sigma(-t)$

$$\mathbb{E}_\varepsilon(\sigma(\varepsilon(x_1) - \varepsilon(x_2))) = 1 - \mathbb{E}_\varepsilon(\sigma(\varepsilon(x_2) - \varepsilon(x_1))) = 1 - \mathbb{E}_\varepsilon(\sigma(\varepsilon(x_1) - \varepsilon(x_2))) \,,$$

so that $g(0) = \mathbb{E}_\varepsilon(\sigma(\varepsilon(x_1) - \varepsilon(x_2))) = 1/2$.

Thus, for $t \leq 0, g(t) \leq 1/2$ while for $t > 0, g(t) > 1/2$. In other words,

$$\text{sign}(t) = \text{sign}\left(g(t) - \frac{1}{2}\right)$$

The result follows by taking $t = \bar{u}(x_1) - \bar{u}(x_2)$. $\qquad\square$

Proposition 2 shows that ordinal preferences induced by $\bar{u}$ (i.e.. $\text{sign}\left(\bar{u}(x_1) - \bar{u}(x_2)\right)$) are in correspondence with the sign of $2\mathbb{E}[Y|X_1 = x_1, X_2 = x_2] - 1$. This motivates the **Sign estimator** that maximizes the agreement between the signs of $\hat{u}^{\text{Sign}}(x_1) - \hat{u}^{\text{Sign}}(x_2)$ and $2Y - 1$:

$$\hat{u}^{\text{Sign}} \in \arg\min_{\tilde{u} \in \mathcal{U}} \mathcal{L}_{0-1}(\tilde{u}) = -\mathbb{E}_{X_1, X_2, Y}\Big[(2Y - 1)\text{sign}(\tilde{u}(X_1) - \tilde{u}(X_2))\Big] \tag{2}$$

The remainder of this Section establishes what the (population level) sign estimator above recovers; the next Section will then study its empirical counterpart.

### 3.1 Ordinal Consistency for General Utilities

We now show that all minimizers of the 0-1 loss defining the Sign estimator are ordinally consistent with $\bar{u}$. Recall that our data generating process assumes $(X_1, X_2)$ is drawn from $\mathcal{X}^2$ according to an exchangable distribution $\mu$. We have:

**Theorem 1.** *Under symmetry, (i.e. Assumption 1), we have:*

$$\bar{u}(X_1) \geq \bar{u}(X_2) \implies \hat{u}^{\mathrm{Sign}}(X_1) \geq \hat{u}^{\mathrm{Sign}}(X_2) \ \mu \ a.s.$$

*Proof.* First, we use Proposition 2 to show that $\bar{u}$ is one of the minimizers of the loss. Denote $\mathrm{ch}(X_1, X_2) = 2(\mathbb{P}(Y = 1 | X_1 = x_1, X_2 = x_2) - \frac{1}{2})$. Then, for an arbitrary function $u'$, we have that:

$$\mathcal{L}_{0-1}(u') - \mathcal{L}_{0-1}(\bar{u}) = \mathbb{E}_{X_1, X_2}[\mathrm{ch}(X_1, X_2)(\mathrm{sign}(\bar{u}(X_1) - \bar{u}(X_2)) - \mathrm{sign}(u'(X_1) - u'(X_2)))]$$

From Proposition 2 we know that $\mathrm{sign}(\bar{u}(X_1) - \bar{u}(X_2)) = \mathrm{sign}(\mathrm{ch}(X_1, X_2))$. Thus:

$$\begin{aligned}
&\mathcal{L}_{0-1}(u') - \mathcal{L}_{0-1}(\bar{u}) \\
&= \mathbb{E}_{X_1, X_2}\left[|\mathrm{ch}(X_1, X_2)|(1 - \mathrm{sign}(u'(X_1) - u'(X_2))\,\mathrm{sign}(\bar{u}(X_1) - \bar{u}(X_2)))\right] \\
&= 2\mathbb{E}_{X_1, X_2}\left[|\mathrm{ch}(X_1, X_2)|\mathbf{1}(\mathrm{sign}(u'(X_1) - u'(X_2)) \neq \mathrm{sign}(\bar{u}(X_1) - \bar{u}(X_2)))\right] \\
&\geq 0
\end{aligned}$$

Hence, $\bar{u}$ is a minimizer of the loss. Next, let $\hat{u}^{\mathrm{Sign}} \in \arg\min_{u'} \mathcal{L}_{0-1}(u')$ be a minimizer of the loss so that $\mathcal{L}_{0-1}(\hat{u}^{\mathrm{Sign}}) - \mathcal{L}_{0-1}(\bar{u}) = 0$. Observe that $\bar{u}(X_1) > \bar{u}(X_2) \implies \mathrm{ch}(X_1, X_2) \neq 0$. But then from the penultimate display above, we must have $\hat{u}^{\mathrm{Sign}}(X_1) - \hat{u}^{\mathrm{Sign}}(X_2) = \mathrm{sign}(\bar{u}(X_1) - \bar{u}(X_2)) \ \mu$ a.s., so that $\hat{u}^{\mathrm{Sign}}(X_1) > \hat{u}^{\mathrm{Sign}}(X_2) \ \mu$ a.s. Finally, recall we assumed that ties $u(X_1) = u(X_2)$ occur on a set of $\mu$ measure zero. This completes the proof. $\square$

### 3.2 Consistency for Linear Utilities up to Positive Scaling

In this section, we deduce from Theorem 1 a stronger consistency result for linearly parametrized models, i.e., where $u(\cdot; \beta) = \langle \phi(\cdot), \beta \rangle$. Here, we define $\varepsilon \triangleq \beta - \bar{\beta}$; the symmetry assumption amounts to $\varepsilon$ and $-\varepsilon$ having the same distribution. We seek to recover $\bar{\beta} = \mathbb{E}_{\beta}[\beta]$ up to positive scaling. That is, we seek to recover $\bar{\mu} \triangleq \bar{\beta}/\|\bar{\beta}\|$. We require a further assumption on the density of $X \triangleq \phi(X_1) - \phi(X_2)$ so that $\bar{\mu}$ is identifiable from pairwise choice data, a pre-requisite for any consistent estimator.

**Assumption 2.** *$X$ is a continuous random variable with density $f_X$. $f_X$ is bounded away from zero in an open set containing the origin.*

Assumption 2 is, to our knowledge, the weakest known sufficient condition that allows for the identification of $\bar{\beta}$; see Fox et al. (2012).[4] In the linear setting, the Sign estimator recovers

$$\hat{\mu}^{\mathrm{Sign}} \in \arg\min_{\theta \in \mathcal{S}^{d-1}} \mathcal{L}_{0-1}(\theta) = -\mathbb{E}_{X_1, X_2, Y}\left[(2Y - 1)\,\mathrm{sign}\left(X^{\top}\theta\right)\right].$$

We then have:

**Corollary 1.** *Under Assumption 1, $\bar{\mu}$ is a minimizer of $\mathcal{L}_{0-1}$ over $\mathcal{S}^{d-1}$. If Assumption 2 also holds, this minimizer is unique so that $\hat{\mu}^{\mathrm{Sign}} = \bar{\mu}$.*

## 4 Finite-Sample Convergence

In this section, we consider the empirical counterpart of the 0-1 loss and derive a sample complexity result. We focus on the linear utility setting and establish cube-root convergence of the Sign estimator to the learning target $\bar{\mu}$.

---

[4]While Fox et al. (2012) address identification through a constructive proof, their approach does not provide a efficient estimator.

**Theorem** (Informal). *Under Assumption 1 (symmetry of $\varepsilon$) and mild regularity on the distributions of $\beta$ and $X$, the Sign estimator achieves a $\tilde{O}(n^{-1/3})$ rate for the empirical risk with high probability.*[5]

To formalize this claim, we define the empirical loss as $\mathcal{L}_{0-1}^n(\theta) \triangleq -\frac{1}{n} \cdot (\sum_{i=1}^n (2Y_i - 1) \cdot \text{sign}(X_i^\top \theta))$, and the corresponding Sign estimator $\hat{\mu}_n^{\text{Sign}} \in \arg\min_{\theta \in \mathcal{S}^{d-1}} \mathcal{L}_{0-1}^n(\theta)$. Notice that the empirical loss is piecewise constant, making the Sign estimator highly degenerate. Nonetheless, as a first step, we establish that any arbitrary family of minimizers $\{\hat{\mu}_n^{\text{Sign}}\}_{n \geq 1}$ of the empirical loss is consistent.

**Proposition 3.** *Under Assumptions 1 and 2, the Sign estimator is consistent, i.e., we have $\mathbb{P}(\lim_{n \to +\infty} \hat{\mu}_n^{\text{Sign}} = \bar{\mu}) = 1$.*

We next turn to a finer sample complexity analysis. We seek to upper bound the angular loss $\hat{\alpha}_n \triangleq \alpha(\hat{\mu}_n^{\text{Sign}}, \bar{\mu})$ between $\hat{\mu}_n^{\text{Sign}}$ and $\bar{\mu}$ with high probability. We focus on the angular loss as it is easy to interpret when comparing two unit vectors. The angle is connected to the $\ell_2$-risk through the identity $\|\theta - \theta'\|^2 = 2(1 - \cos(\alpha(\theta, \theta')))$.

We make use of two assumptions to establish local strong convexity of the population risk. First, we require $X$ to have bounded density, bounded support, and have density bounded away from zero in a neighborhood of the origin, strengthening Assumption 2.[6]

**Assumption 3.** *Suppose that $X$ is a continuous random variable with density $f_X$ with respect to the Lebesgue measure. Moreover, there exist $R, r, C_X, \rho_X > 0$ such that $\forall x \in \mathcal{X}$: $f_X(x) \leq C_X$, $\|X\| \leq R$ almost surely, and $\forall x \in B(0, r) : f_X(x) \geq \rho_X$.*

Additionally, we require a uniform central-mass condition: every projection of the noise vector $\varepsilon = \beta - \bar{\beta}$ places at least a fixed probability on a small neighborhood of zero.

**Assumption 4.** *Suppose that we have $\rho_\varepsilon \triangleq \inf_{u \in \mathcal{S}^{d-1}} \mathbb{P}_\varepsilon(|(u^\top \varepsilon| \leq \frac{\|\bar{\beta}\|}{2\sqrt{2}}) > 0$.*

This assumption is easily satisfied; for example, it holds for Gaussian mixed logits.

The next theorem states our finite-sample bound on the angular loss between $\hat{\mu}_n^{\text{Sign}}$ and $\bar{\mu}$, yielding a cube-root convergence rate.

**Theorem 2.** *Fix $\alpha_0 \in [0, \frac{\pi}{4}]$ and $\delta \in (0, \frac{1}{2})$. Define $n_0$ as the universal constant such that with probability at least $1 - \delta$, we have $\alpha(\hat{\mu}_n^{\text{Sign}}, \bar{\mu}) \leq \alpha_0$ for every $n \geq n_0$. If Assumptions 1, 3, and 4 hold, then there exist $K, C_0 > 0$ independent of $n, \delta$ such that with probability at least $1 - 2\delta$, for every $n \geq n_0$*

$$\alpha(\hat{\mu}_n^{\text{Sign}}, \bar{\mu}) \leq \left(\frac{K}{C_0}\right)^{\frac{2}{3}} \left(\frac{d - 1 + \log(\frac{\log(n)}{\delta})}{n}\right)^{\frac{1}{3}} + \sqrt{\frac{\log(\frac{\log(n)}{\delta})}{n}} \ . \tag{3}$$

*Specifically, we have $K = \Theta(\log(\frac{A^d}{C_X Vol(B_{d-2}(R))R^2})\sqrt{d} + \frac{2C_X Vol(B_{d-2}(R))R^2}{\sqrt{d}})$ where $A$ is a universal constant and $C_0 = \Theta(\|\bar{\beta}\|\sigma'(\frac{\|\bar{\beta}\|r}{\sqrt{2}})\rho_\varepsilon \rho_X Vol(B_{d-2}(r))r^2)$.*

We further illustrate how this rate depends on the dimension $d$ when instantiated for specific distributions, for simplicity, we present this dimension analysis for a uniform feature distribution and Gaussian preference vectors.

**Corollary 2.** *Suppose that $X$ follows a uniform distribution in $B(0, 1)$ and $\beta \sim \mathcal{N}(\bar{\beta}, \frac{1}{d}I_d)$. Then, $K = \Theta(d^{\frac{3}{2}})$, $C_0 = \Theta(d)$ and the upper bound on the angular loss in equation 3 amounts to*

$$\alpha(\hat{\mu}_n^{\text{Sign}}, \bar{\mu}) \lesssim \left(\frac{d^2}{n}\right)^{\frac{1}{3}}$$

---

[5]We use $\tilde{O}(\cdot)$ to hide additional factors with poly-logarithmic dependence in $n$ and the failure probability $\delta$, and dependence on other instance parameters including $d$. See Theorem 2.

[6]We remark that our analysis can handle a standard sub-gaussian tail assumption (instead of boundedness) but it degrades our sample complexity bound by instance-dependent factors.

A key step in our analysis is to establish local curvature of the population risk around $\bar{\mu}$; the proof ideas may be of independent interest. Our analysis uses a localized law of large numbers to control the empirical excess risk and obtain sharp finite sample bounds.

To our knowledge, this is among the first finite-sample recovery guarantees for the aggregate parameter $\bar{\mu}$ in general mixtures of generalized linear models. In contrast, the nonparametric random-coefficients literature provides error rates where the exponent typically deteriorates with the dimension $d$ (e.g., Gautier & Kitamura (2013); Fox et al. (2016)). While polynomial sample complexity rates are achievable for finite mixtures, they typically require linear independence among the utility vectors (Ammar et al., 2014; Oh & Shah, 2014; Sedghi et al., 2016), rendering them inapplicable in our setting. Of course, the improvement is possible because our estimator sidesteps the need to learn the mixture components, directly targeting $\bar{\mu}$ by exploiting the symmetry structure.

**Computational implementation.** Despite the conceptual simplicity of our estimator, the 0-1 loss is non-differentiable and NP-hard in general (Ben-David et al., 2003). That said, binary classification with 0-1 loss is a widely studied learning task, with practical implementations using convex surrogates, smooth relaxations of the sign function (Bartlett et al., 2006; Bao et al., 2020), or integer programming. Our implementations approximates the sign function in the 0-1 loss through the simple point-wise smooth function $\text{sign}(t) \approx 2\sigma(\lambda t) - 1$ where $\lambda$ is a temperature parameter that we anneal during the training.

Importantly, while using a convex surrogate function (i.e., logistic or hinge loss) may seem natural, it would essentially revert to the RLHF estimator (which uses logistic loss). The use of convex surrogates is typically justified under the assumption *classification calibration*— that the model class is realizable (Bartlett et al., 2006; Bach, 2024)—an assumption that clearly fails in our setting since heterogeneity causes the RLHF estimator to be misspecified.

## 5 EXPERIMENTS

This section conducts an empirical study of the Sign estimator. We test the predictive accuracy of our estimator against existing RLHF methods in preference modeling tasks, on two standard benchmarks (SPH and the Anthropic Helpfulness training datasets) used in previous literature.

Nonetheless, existing empirical benchmarks present two limitations: (1) the ground truth preferences are unknown, so we cannot directly measure the learned preference model's alignment with the utilitarian aggregate; (2) heterogeneity levels are not measurable and may not reflect real-world deployments (e.g., LLM user population is more diverse). Therefore, we propose a new semi-synthetic methodology. The data for our study is itself of note: it consists of preferences obtained from a representative panel of 200 synthetic personas across 43,834 pairs of alternative answers. The questions and alternatives are drawn from the Anthropic Helpfulness training corpus Bai et al. (2022a), while the synthetic personas were carefully selected from a set of over 2,500 'digital twins' curated by Toubia et al. (2025). Each digital twin is constructed from an approximately 500-question interview with a real human interviewee. Our main findings are the following:

**Estimation error:** The RLHF estimator has significant bias. This bias is strongly mitigated by the Sign estimator: angular estimation error with the true mean utility vector is reduced by about 35% (from 63° to 41°); while disagreement rates with optimal choices under the true mean utility reduce by about 40%. Note that disagreement rates corresponds to one minus classification accuracy.[7]

**Comparative Statics:** Since our real world dataset of alterative answers itself limits observed heterogeneity, we artificially increase heterogeneity in the experiments above, and show that this only serves to increase the relative improvement of the Sign estimator.

---

[7]This highlights the value of measuring disagreement/accuracy against $\text{sign}(\bar{\beta}^T X)$ instead of $\text{sign}(Y)$. Considering an empirical set-up where $\bar{\beta}$ is known allows us to measure performance with respect to the ground truth utility, whereas the metric of classification accuracy factors in residual noise from randomness in preferences.

**Panel Data Heuristics:** Panel data heuristics would seek to leverage the identity of users (PII) to learn mean utility: data not used by the Sign estimator or typical reward learning pipelines. As a representative of this class, we implement an EM algorithm proposed recently for reward learning. Despite its simplicity (a drop-in replacement for CE loss in a typical reward learning pipeline) and not using PII, the Sign estimator dominates.

**Set-up:** Our alternatives $\mathcal{X}$ consist of $43,834$ prompt-completions pairs from Anthropic's Helpfulness training corpus (Bai et al., 2022a). Our population consists of 200 *personas* chosen from a set of $2,500$ digital twins (Toubia et al., 2025), which reproduce the economic behaviors and preferences of real individuals from a representative US panel. Each persona consists of answers to a battery of 500 questions from a distinct human interviewee. Toubia et al. (2025) showed that when prompted with a subset of these answers, language models could predict the interviewee's responses to held-out questions with approximately 88% accuracy; Park et al. (2024b) report similar findings. For each persona and each pair $(x_1, x_2) \in \mathcal{X}$, we prompt GPT-4o-mini with that persona's survey summary (see Appendix J) to extract the persona's exact probability of preferring $x_1$. The dataset exhibits interesting and realistic heterogeneity: Figure 3 (left panel) shows a histogram of difference in preferences across a randomly chosen prompt and pair of personas; the mean difference is 10%. The right panel shows self-reported demographic features of the 200 corresponding individuals. Additionaly, in table 1, we run our models on the true labels of the Helpfulness Dataset, as well as on a subset of 150,000 data points of the Stanford Human Preferences Dataset Ethayarajh et al. (2022).

**Ground-truth Utilities:** Since we have access to the actual preference probabilities—rather than merely sampled preferences—for each persona, we can effectively recover each persona's implicit utility vector up to the choice of origin. We choose the function class $u(x; \beta^k) = \phi(x)^\top \beta^k$ where the feature map $\phi(\cdot) \in \mathbb{R}^{2880}$ represents features from `gpt-oss-20b`'s final hidden layer and $\beta^k \in \mathbb{R}^{2880}$ is persona $k$'s utility vector. This representation provides a good fit to the observed choice probabilities: the Jensen-Shannon divergence between the probabilities yielded by this linear utility model and the observed probabilities on a holdout set is 0.08 (on a scale of 0 to $\log 2$). In what follows, we take $\mathcal{B} = \{\beta^k\}_{k \in [200]}$ to be the set of these learned utility vectors, and treat the uniform distribution over this set as our ground truth distribution of users.

**Preference Datasets:** Our source dataset $\{(X_1, X_2)_{ij}, Y_{ij} : i \in [n_u], j \in [n_p]\}$ is generated as follows. We sample $n_u = 4000$ users $\beta_1, \ldots, \beta_{n_u}$ uniformly from $\mathcal{B}$ with replacement. For each user $\beta_i$, we draw $n_p = 5$ prompt-completion pairs from $\mathcal{X}$ without replacement, where $(X_1, X_2)_{ij}$ denotes the $j$-th pair shown to user $i$. We then generate the labels $Y_{ij}$ of user $\beta_i$'s preferred alternative in the $j$-th pair.

## 5.1 Results

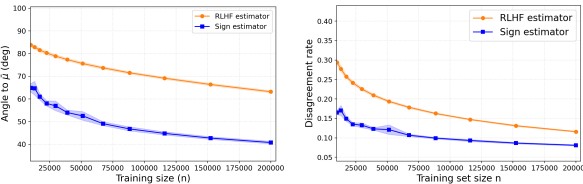

Figure 1: Sign Estimator reduces angular estimaton error and disagreement rates by $\sim 40\%$. Left: Angle error with true mean. Right: Disagreement rate with true mean utility.

**RLHF vs Sign Estimator on Real Preferences:** Figure 1 compares the RLHF and Sign estimators as we vary the train data size $n$. The left panel reports the angular loss $\alpha(\hat{\mu}, \bar{\mu})$ for each estimator $\hat{\mu}$ while the right panel reports the rate of disagreements between $\bar{\mu}$ and the estimator across *all* 43,834 questions in the data. The Sign estimator achieves significant improvement over the RLHF approach, reducing angular error from approximately 63° to 41° for a plausible training size of $n = 200,000$. This translates to an important 40%

reduction in disagreement rates from roughly 13% to less than 8%. We emphasize that the setup here does not satisfy the Assumptions we made for our theoretical development illustrating robustness. As an important set of comparative statics, we conduct additional synthetic experiments where we raise the level of heterogeneity in preferences by scaling $\beta_i - \bar{\beta}$; see Appendix A for details. We find that increasing heterogeneity amplifies the performance gap between the two methods.

**Comparison with EM:** Our methods so far treat all pairwise comparisons independently, ignoring the panel data structure—that each of the $n_u$ users annotates $n_p$ prompt-completion pairs. By pooling all user data, our approach maintains compatibility with standard RLHF pipelines, requiring only a modified loss function rather than architectural changes. In contrast, panel estimation methods leverage individual-level information to explicitly model users' heterogeneous preferences and seek to recover the distribution of $\beta$s. We use an EM algorithm, proposed for RLHF by Chakraborty et al. (2024, Alg. 2) and Poddar et al. (2024). The algorithm alternates between assigning users to one of $K$ 'homogeneous' segments (E-step) and fitting a Bradley-Terry reward model within each segment (M-step); see Appendix G.

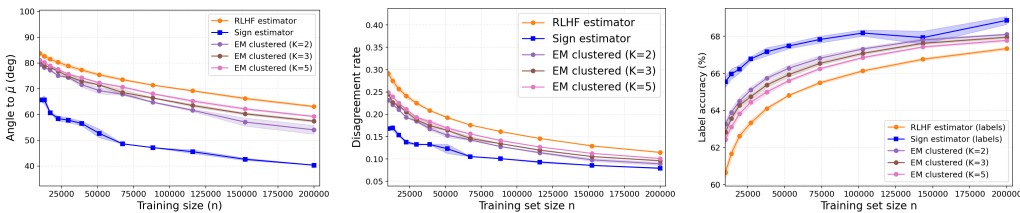

Figure 2: RLHF, EM vs Sign. Angular Estimation error (left) , utilitarian disagreement rate (center), and label accuracy(right).

When $K = 1$, the approach reduces to the standard RLHF estimator $\hat{\beta}^{\mathrm{RLHF}}$, where we pool the data from all personas, whereas $K = 200$ fits one reward model $\hat{\beta}_i$ on each persona $\beta_i$'s preference data. In this section, we explore intermediate values of $K \in \{2, 3, 5\}$. Figure 2 reports our performance metrics in this context. We see that the EM algorithm with $K = 2$ achieves improvements over the RLHF estimator—which can be attributed to better capturing heterogeneity—but performance degrades for $K \in \{3, 5\}$—suggesting a variance counter-effect due to segmentation of the user sample, or possibly convergence of the EM algorithm to spurious stationary points. The Sign estimator still emerges as the most effective method, although it ignores the panel information and maintains a far simpler and less computationally intensive implementation. Table 1 shows the accuracy of the sign estimators in preferring the same outputs as the ground-truth $\bar{\beta}$ (utilitarian accuracy) as well as standard accuracy of the responses preferred by the annotators. We note that the values of standard accuracy are as expected and reported in past work Bai et al. (2022a) since the annotator choices are inherently noisy and the annotators have heterogeneous preferences and low agreement rate, which is the motivation for this work.

| | Sign (ours) | RLHF | EM (K=2) | EM (K=3) | EM (K=5) |
|---|---|---|---|---|---|
| HH-Anthropic(utilitarian accuracy) | **92.00** | 88.51 | 90.91 | 90.43 | 89.88 |
| HH-Anthropic(Persona labels) | **68.86** | 67.33 | 68.10 | 67.94 | 67.77 |
| HH-Anthropic | **68.58** | 66.89 | 65.01 | 63.94 | 64.19 |
| SHP | **68.33** | 66.50 | 67.01 | 60.83 | 61.44 |

Table 1: Comparison of the accuracy of the different reward learning algorithms trained on multiple preference datasets. The first row considers the utilitarian accuracy(Whether the model prefers the same output as the average ground-truth reward model), while the others calculate the standard accuracy against the choices made by the annotators.

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

## A ADDITIONAL EXPERIMENTS

We first note the heterogeneity in the selected population through Figure 3

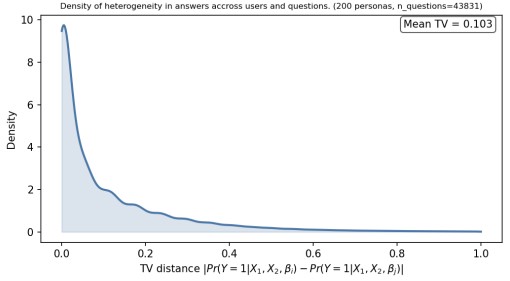

**Persona Demographics**

**Region**: South (40.0%); Midwest (22.5%) . . .
**Gender**: Male (56.0%); Female (44.0%) . . .
**Age**: 50-64 (34.5%); 30-49 (34.0%) . . .
**Education**: College (34.0%); Postgrad (22.5%)
**Race**: White (68.5%); Black (13.0%) . . .
**Marital**: Married (40.0%); Never (33.0%)
**Religion**: Protestant (24%); Catholic (21.5%)
**Income**: $100k (33.0%); $30–50k (20.0%) . . .
**Views**: Moderate (27.5%); Liberal (24.5%) . . .
**Employment**: FT (43.5%); PT (14.0%) . . .

Figure 3: (left) Heterogeneity histogram and (right) Persona Demographics .

We show in Figure 4 and Table 2 the effect of scaling the variance in the users utility functions and its impact on the estimation risk of both the RLHF and Sign estimators.

Table 2: Increased Heterogeneity Increases Relative Improvement: Coefficient of Variation (CV) of $\beta$, angle error, and disagreement for two estimators.

| Scale | CV | $\alpha(\hat{\mu}, \bar{\mu})$ | | Disagreement (%) | | $\delta\%$ |
|---|---|---|---|---|---|---|
| | | RLHF | Sign | RLHF | Sign | |
| 1.0× | 0.59 | 63.24 | **41.55** | 11.60 | **8.21** | 29.22% |
| 4.0× | 1.18 | 65.89 | **43.36** | 12.58 | **8.92** | 29.09% |
| 8.0× | 1.67 | 70.85 | **47.11** | 15.42 | **9.95** | 35.47% |
| 12.0× | 2.05 | 74.36 | **49.68** | 18.21 | **11.2** | 38.49% |

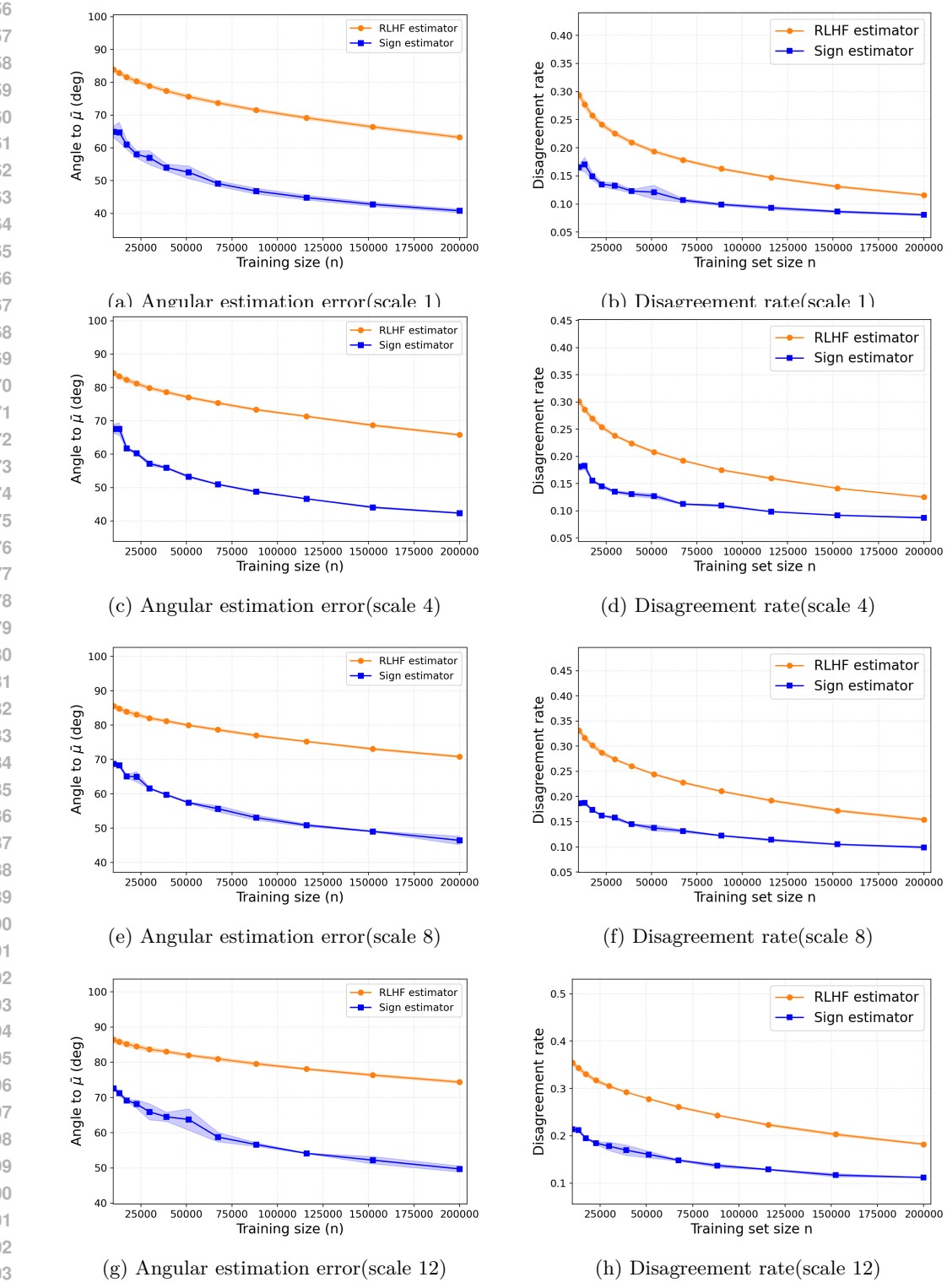

(a) Angular estimation error(scale 1)

(b) Disagreement rate(scale 1)

(c) Angular estimation error(scale 4)

(d) Disagreement rate(scale 4)

(e) Angular estimation error(scale 8)

(f) Disagreement rate(scale 8)

(g) Angular estimation error(scale 12)

(h) Disagreement rate(scale 12)

Figure 4: Comparison of 4 levels of scale (1,4,8,12) and the effect of introducing more heterogeneity in estimating the direction of the mean. Predictably, a higher variance amplifies the estimation error of both estimators, while the Sign estimator seems to handle it more gracefully.

# B BACKGROUND

**Utilitarian theorem.**   A classic result due to Harsanyi (1955) shows that if (i) each user's preferences over stochastic prospects admit a von Neumann-Morgenstern-Marschak (VNM) utility representation, (ii) the social preferences satisfy the VNM axioms and treat users symmetrically, and (iv) individual-level indifference between two prospects implies a similar indifference for social preferences, then the only compatible social welfare functional is (up to a positive affine transformation) the sum of individual utilities.

Thus, utilitarian aggregation is justified under widely accepted axioms. In the context of RLHF, this motivates the approach that consists of learning a 'reward function' that aggregates users' utilities additively: maximizing expected social welfare reduces to maximizing the expected sum of user-level utilities.[8]

**RLHF pipeline and learning task.**   In the standard RLHF procedure, we are given a reference policy $\pi^{\mathrm{ref}}(\cdot)$, describing the language model random outputs, i.e., given prompt-completion pair $x = (p, c) \in \mathcal{X}$, $\pi^{\mathrm{ref}}(x)$ is the probability of generating completion $c$ conditional to prompt $p$. In the remainder of this section, we omit the reference to the prompt $p$ in our notion of policy (i.e., all subsequent claims are conditional to each prompt).

The fine-tuned policy (under utilitarian aggregation) chooses a generative distribution over completions $\pi \in \Delta^{\mathcal{X}}$ to maximize the user expected utility plus a KL-divergence penalty relative to the reference policy. That is, $\pi^{\mathrm{RLHF}}(\cdot) \in \arg\max_{\pi \in \Delta^{\mathcal{X}}} \Phi_{\bar{u}, \gamma}(\pi)$ where

$$\Phi_{\bar{u}, \gamma}(\pi) \triangleq \mathbb{E}_{x \sim \pi}\left[\bar{u}(x)\right] - \gamma D_{KL}\left(\pi \| \pi^{\mathrm{ref}}\right) = \mathbb{E}_{\beta, x \sim \pi}\left[u(x; \beta)\right] - \gamma D_{KL}\left(\pi \| \pi^{\mathrm{ref}}(\cdot)\right) ,$$

and $\gamma > 0$ is the so-known temperature parameter. This formulation justifies our learning target $\bar{u}(\cdot) = \mathbb{E}_{\beta}[u(\cdot; \beta)]$—and $\bar{\beta} = \mathbb{E}[\beta]$ in the special case of linear utility—as a sufficient statistic for policy fine-tuning. The target $\bar{u}(\cdot)$ is our notion of reward model for a heterogeneous population of users.

We say that two reward/utility functions $u(\cdot), u'(\cdot)$ are *ordinally consistent* if $u(x_1) \geq u(x_2) \Leftrightarrow u'(x_1) \geq u'(x_2)$ for all pairs of alternatives $x_1, x_2 \in \mathcal{X}$. That is, they induce the same preferences over alternatives, i.e., there exists a monotone mapping $f : \mathbb{R} \to \mathbb{R}$ such that $u'(x) = f(u(x))$ for all $x \in \mathcal{X}$. For linear utility functions with utility vectors $\beta, \beta'$, them being ordinally consistent amounts to both vectors sharing the same direction (if $\mathcal{X}$ is full rank), i.e., $\beta' = \lambda\beta$ for some scalar $\lambda > 0$.

In this work, we will be satisfied with recovering $\bar{u}$ up to induced preferences—or $\bar{\beta}$ up to a positive scaling in the linear case. We notice that this learning target suffices to determine the class of RLHF policies or a close approximation thereof. Define $\Pi_{\bar{u}, \gamma}^{\mathrm{RLHF}} = \arg\max_{\pi} \Phi_{\bar{u}, \gamma}(\pi)$ and $\Pi_{\bar{u}}^{\mathrm{RLHF}} = \bigcup_{\gamma \geq 0} \Pi_{\bar{u}, \gamma}^{\mathrm{RLHF}}$, then

- In the linear case, $\Pi_{\beta}^{\mathrm{RLHF}} = \Pi_{\beta'}^{\mathrm{RLHF}}$ for any two utility vectors $\beta, \beta'$ sharing the same direction, and in particular, $\Pi_{\bar{\beta}}^{\mathrm{RLHF}} = \Pi_{\bar{\mu}}^{\mathrm{RLHF}}$ where we recall that $\bar{\mu} = \bar{\beta}/\|\bar{\beta}\|$. Therefore, focusing on the learning target $\bar{\mu}$ suffices to fully determine the class of fine-tuned policies—up to temperature rescaling.

- For general utility functions, $\Pi_{u, 0}^{\mathrm{RLHF}} = \Pi_{u', 0}^{\mathrm{RLHF}}$ for any two ordinally consistent utility functions $u, u'$. That is, when the temperature is zero, and the models have deterministic outputs, ordinal consistency is also without loss.

- However, it is generally not true that $u, u'$ being ordinally consistent suffices for $\Pi_{u}^{\mathrm{RLHF}} = \Pi_{u'}^{\mathrm{RLHF}}$. In this context, we recover ordinal consistency over policies: when the reference policy $\pi^{\mathrm{ref}}$ is itself ordinally consistent with $u, u'$, then the fine-tuned policies obtained from the utility functions $u$ and $u'$ are ordinally consistent.

---

[8]We note, however, that our notion of utility is over deterministic outcomes, rather than stochastic prospects (lotteries), as in the utilitarian theorem.

## C  Failure of the RLHF Estimator: Two Class Example

Recall we have two types of users, $\mathcal{B} = \{1, 2\}$, faced with two alternatives, $\mathcal{X} = \{o, 1\}$ where users of type 1 make up a fraction $\alpha$ of the population. Further, we have $u(1; 1) = 1$ while $u(1; 2) = -M < 0$. Observe that $\bar{u}(1) = \alpha + (1 - \alpha)(-M)$. Consequently, $\bar{u} < 0 \iff \alpha < M/1 + M$.

Next observe that since $\sigma(\tilde{u}^{\text{naive}}(1)) = \mathbb{P}(Y = 1) = \alpha\sigma(1) + (1 - \alpha)\sigma(-M)$, we have that

$$\tilde{u}^{\text{naive}}(1) > 0 \iff \alpha > \frac{\sigma(M) - 1/2}{\sigma(1) + \sigma(M) - 1} \uparrow \frac{1}{2\sigma(1)} < 0.7$$

It follows that for any $M \geq 3$ and $0.7 \leq \alpha \leq 1 - 1/M < M/(1 + M)$, we have $\tilde{u}^{\text{naive}}(1) > 0$ while $\bar{u}(1) < 0$.

## D  Analysis of the Naive Estimator for Gaussian Data

### D.1  Analysis of MLE's bias and sufficient condition for recovery

What stronger assumptions makes the Naive estimator consistent under Gaussian data? The next proposition identifies a sufficient condition—*conditional symmetry* in the noise $\varepsilon = \beta - \bar{\beta}$. This condition, however, is quite stringent—e.g., in the case of the mixed logit model $\beta \sim \mathcal{N}(\bar{\beta}, \Sigma)$, it essentially requires $\Sigma$ to be a diagonal matrix. The proof of Proposition 4 is instructive as it isolate the source of asymptotic bias in the Naive estimator.

**Proposition 4.** *Suppose that $X \sim \mathcal{N}(0, I_d)$. Letting $\varepsilon = \beta - \bar{\beta}$ denote the mean-centered noise in the utility vector, we denote by $\epsilon_\| := (\varepsilon^\top \bar{\mu})\bar{\mu}$ its random component parallel to $\bar{\mu}$ and by $\epsilon_\perp := \varepsilon - \epsilon_\|$ its orthogonal component. If the distribution of $\epsilon_\perp$ conditional on $\epsilon_\|$ is symmetric, i.e., $\mathbb{P}(\epsilon_\perp \in \cdot | \epsilon_\|) = \mathbb{P}(-\epsilon_\perp \in \cdot | \epsilon_\|)$, then the Naive estimator consistently recovers the direction $\bar{\mu}$.*

Suppose $X \sim N(0, I_d)$. Then using Stein's lemma on both sides of the MLE equation, we get that:

$$E_X(\sigma'(X^T\hat{\beta}))\hat{\beta}$$
$$= E_\epsilon(E_X(\sigma'(X^T(\bar{\beta} + \epsilon))(\bar{\beta} + \epsilon))$$
$$= E_\epsilon(E_X(\sigma'(X^T(\bar{\beta} + \epsilon)))\bar{\beta} + E_\epsilon(E_X(\sigma'(X^T(\bar{\beta} + \epsilon))\epsilon_\|) + E_\epsilon(E_X(\sigma'(X^T(\bar{\beta} + \epsilon))\epsilon_\perp)$$

The first two components are in the (positive) direction of $u$, then, we have that the MLE recovers the direction of u if and only if:

$$E_\epsilon(E_X(\sigma'(X^T(\bar{\beta} + \epsilon))\epsilon_\perp) = 0 \tag{4}$$

We will show that a sufficient condition is that conditionally on the value of $\epsilon_\|$, $\epsilon_\perp$ is symmetric, that is $\epsilon_\perp|_{\epsilon_\|} \stackrel{d}{=} -\epsilon_\perp|_{\epsilon_\|}$. But in general, just symmetry of $\epsilon_\perp$ is not sufficient by providing a counterexample.

If $\epsilon_\perp|_{\epsilon_\|} \stackrel{d}{=} -\epsilon_\perp|_{\epsilon_\|}$. Then we have:

$$E_\epsilon(E_X(\sigma'(X^T(\bar{\beta} + \epsilon))\epsilon_\perp) = E_\epsilon(E_X(\sigma'(X^T(\bar{\beta} + \epsilon_\| + \epsilon_\perp))\epsilon_\perp)$$
$$= E_\epsilon(E_X(\sigma'(X^T(-\bar{\beta} - \epsilon_\| + \epsilon_\perp))\epsilon_\perp)$$
$$= E_\epsilon(E_X(\sigma'(X^T(-\bar{\beta} - \epsilon_\| - \epsilon_\perp))(-\epsilon_\perp))$$
$$= -E_\epsilon(E_X(\sigma'(X^T(\bar{\beta} + \epsilon_\| + \epsilon_\perp))\epsilon_\perp) .$$

Thus, $E_\epsilon(E_X(\sigma'(X^T(\bar{\beta} + \epsilon))\epsilon_\perp) = 0$. The first line comes from just the decomposition of $\epsilon$. The second line comes from the ability of a standard gaussian to flip a direction while keeping others intact. That is for two orthogonal vectors u and v, $(X^T u, X^T v) \stackrel{d}{=} (-X^T u, X^T v)$. The third line comes from the assumption $\epsilon_\perp|_{\epsilon_\|} \stackrel{d}{=} -\epsilon_\perp|_{\epsilon_\|}$. The fourth line is a consequence of the evenness of the function $\sigma'$.

### D.2 Counter-example

In this section, we construct a simple counter-example in the linear utility case showing that the Naive estimator fails to recover ordinal preferences under symmetry (Assumption 1).

Let $d = 2$, $X \sim N(0, I_2)$, $\bar{\beta} = (1, 0)$ and

$$\epsilon = \begin{cases} (1, -1) & \text{with probability } 0.5 \\ (-1, 1) & \text{with probability } 0.5 \end{cases}$$

Then $\epsilon$ is symmetric, but we see that conditionally on $\epsilon_\parallel$ which gives the first coordinate, $\epsilon_\perp$ is given by the second coordinate which is deterministic and is non-zero, hence not symmetric.

We check that the condition $E_\epsilon(E_X(\sigma'(X^T(\bar{\beta} + \epsilon))\epsilon_\perp) = 0$ is not satisfied. We have:

$$E_\epsilon(E_X(\sigma'(X^T(\bar{\beta} + \epsilon))\epsilon_\perp) = \frac{1}{2}E_X(\sigma'(X^T[\begin{pmatrix} 1 \\ 0 \end{pmatrix} + \begin{pmatrix} 1 \\ -1 \end{pmatrix}])) \begin{pmatrix} 0 \\ -1 \end{pmatrix} + \frac{1}{2}E_X(\sigma'(X^T[\begin{pmatrix} 1 \\ 0 \end{pmatrix} + \begin{pmatrix} -1 \\ 1 \end{pmatrix}])) \begin{pmatrix} 0 \\ 1 \end{pmatrix}$$

$$= \frac{1}{2}[E_X(\sigma'(2X_1 - X_2)) - E_X(\sigma'(X_2))]$$

$$= \frac{1}{2}[E_{Z \sim N(0,5)}(\sigma'(Z)) - E_{Z \sim N(0,1)}(\sigma'(Z))]$$

$$\neq 0$$

Hence, $\hat{\beta}$ is not proportionnal to $\bar{\beta}$.

## E    Proof of Theorem 2

The first step of proving non asymptotic rates of convergence is to lower bound the excess risk $\mathcal{L}(\theta) - \mathcal{L}(\bar{\mu})$ by the angle $\alpha(\theta, \bar{\mu})$. This is achieved when $\alpha(\theta, \bar{\mu}) \leq \frac{\pi}{4}$.

Let $\alpha_0 \leq \frac{\pi}{4}$ be a fixed angle value and let $\delta > 0$.

We therefore use the corollary to get the universal $n_0$ such that with probability at least $1 - \delta$: $\forall n \geq n_0 : \hat{\alpha}_n \leq \alpha_0$. In the rest of the paper, we condition on this event and suppose $n \geq n_0$. The following theorem provides a curvature inequality that is a quadratic lower bound of the excess risk by the square of the angle.

**Theorem 3.** *Let $\theta \in S$ such that $\alpha(\theta, \bar{\mu}) \leq \alpha_0$. Suppose $X$ satisfies the local mass assumption: $\exists r \in [0, 1] : \rho_X \triangleq \inf_{x \in B(0,r)} f_X(x) > 0$. Suppose the noise $\varepsilon$ satisfies the local mass assumption $\rho_\varepsilon = \mathbb{P}_\varepsilon(||\varepsilon|| \leq \frac{||\bar{\beta}||}{2\sqrt{2}}) > 0$.*

*Then, there exists a constant $C_0$ that depends on $||\bar{\beta}||, \rho_X$ and $\rho_\varepsilon$ such that for $\alpha \triangleq \alpha(\theta, \bar{\mu})$:*

$$C_0\alpha(\theta, \bar{\mu})^2 \leq \mathcal{L}(\theta) - \mathcal{L}(\bar{\mu}) \tag{5}$$

*Proof.* We have:

$$\mathcal{L}(\theta) - \mathcal{L}(\bar{\mu}) = E_X(|ch(X)|\mathbb{1}(\text{sign}(X^\top\theta) \neq \text{sign}(X^\top\bar{\mu}))) ,$$

with $ch(X) = 2E_\varepsilon(\sigma(X^T(\bar{\beta} + \varepsilon))) - 1$. To control the curvature of $ch(X)$, we need to truncate the tails of $X$ and $\varepsilon$ where the sigmoid $\sigma$ saturates. Let $a = \frac{||\bar{\beta}||r}{\sqrt{2}}, b = \frac{r}{2\sqrt{2}}$. When

$||X|| \leq r$ and $|X^\top \bar\beta| \leq \frac{a}{2}$, we have:

$$\forall t \in \left[-\frac{a}{2}, \frac{a}{2}\right] : \mathbb{E}(\sigma'(t + X^T \varepsilon)) \geq \mathbb{E}_\varepsilon(\sigma'(\frac{a}{2} + |X^\top \varepsilon|))$$

$$\geq E_\varepsilon(\sigma'(\frac{a}{2} + |X^\top \varepsilon|)\mathbb{1}(|X^\top \varepsilon| \leq \frac{a}{2}))$$

$$\geq \sigma'(a)\mathbb{P}_\varepsilon(|X^\top \varepsilon| \leq \frac{a}{2})$$

$$\geq \sigma'(a)\mathbb{P}_\varepsilon(|(\frac{1}{||X||}X)^\top \varepsilon| \leq \frac{a}{2r})$$

$$\geq \sigma'(a) \underbrace{\inf_{u \in S} \mathbb{P}_\varepsilon(|(u^\top \varepsilon| \leq \frac{||\bar\beta||}{2\sqrt{2}})}_{\rho_\varepsilon}$$

$$\geq \sigma'(a)\rho_\varepsilon .$$

Since $\sigma'$ is bounded, we can apply the Dominated Convergence theorem and integrate from 0 to $X^\top \bar\beta$ to get when $||X|| \leq r$ and $|X^\top \bar\beta| \leq \frac{a}{2}$:

$$|ch(X)| \geq 2|X^\top \bar\beta|\sigma'(a)\rho_\varepsilon .$$

From this we can lower bound the excess risk by:

$$\mathcal{L}(\theta) - \mathcal{L}(\bar\mu) \geq 2\sigma'(a)\rho_\varepsilon|E_X[|X^\top \bar\beta|\mathbb{1}(\text{sign}(X^\top \theta) \neq \text{sign}(X^\top \bar\mu), |X^\top \bar\beta| \leq \frac{a}{2}, ||X|| \leq r)]$$

$$\geq 4\sigma'(a)\rho_\varepsilon|E_X[|X^\top \bar\beta|\mathbb{1}(X^\top \theta < 0, X^\top \bar\mu > 0, |X^\top \bar\mu| \leq \frac{a}{2||\bar\beta||}, ||X|| \leq r)] .$$

The second inequality is obtained by the symmetry of X.

We next consider the plan $span(\{\theta, \bar\mu\})$ and let $v \in span(\{\theta, \bar\mu\})$ such that $v \perp \bar\mu$ and oriented such that $0 \leq \alpha \leq \alpha_0$.

Let the set $A \triangleq \{x \in \mathbb{R}^d / ||x|| \leq r, 0 < x^T \bar\mu \leq tan(\alpha)b, -b \leq X^\top v < -(\alpha)X^\top \bar\mu\}$. We will show that this is a portion of the disagreement set. For $x \in A$, we have:

$$x^\top \mu \leq \tan(\alpha)b$$
$$\leq b \qquad\qquad \text{because } \alpha \leq \alpha_0 \leq \frac{\pi}{4}$$
$$\leq \frac{a}{2||\bar\beta||} .$$

And $||x|| \leq r, x^\top \bar\mu > 0$. We also have:

$$x^\top \theta = \cos(\alpha)x^\top \bar\mu + \sin(\alpha)x^\top v$$
$$< \cos(\alpha)(x^\top \bar\mu - \tan(\alpha)\text{cotan}(\alpha)x^\top \bar\mu)$$
$$< 0 .$$

Hence $A \subset \{x \in \mathbb{R}^d / x^\top \theta < 0, x^\top \bar\mu > 0, |x^\top \bar\mu| \leq \frac{a}{2||\bar\beta||}, ||x|| \leq r\}$. We can now lower bound the excess risk by:

$$\mathcal{L}(\theta) - \mathcal{L}(\bar\mu) \geq 4\sigma'(a)\rho_\varepsilon|E_X[|X^\top \bar\beta|\mathbb{1}(X \in A)]$$

On the other hand, from the local density assumption on X, there exists a minimal density $\rho'_X$ for the projection of X on $(\{\theta, \bar{\mu}\})$. Hence:

$$\mathcal{L}(\theta) - \mathcal{L}(\bar{\mu}) \geq 4\sigma'(a)\rho_\varepsilon ||\bar{\beta}|| |E_X[|X^\top \bar{\mu}| \mathbb{1}(X \in A)]$$

$$\geq 4\sigma'(a)\rho_\varepsilon ||\bar{\beta}|| \int_0^{tan(\alpha)b} \int_{(\alpha)t}^b \rho'_X t \, ds \, dt$$

$$\geq 4\sigma'(a)\rho_\varepsilon ||\bar{\beta}|| \rho'_X \int_0^{(\alpha)b} t(b - (\alpha)t) dt$$

$$\geq 4\sigma'(a)\rho_\varepsilon ||\bar{\beta}|| \rho'_X (\frac{b^2}{2}\tan^2(\alpha) - (\alpha)\frac{\tan^3(\alpha)b^3}{3})$$

$$\geq 4\sigma'(a)\rho_\varepsilon ||\bar{\beta}|| \rho'_X \frac{b^2}{2}\tan^2(\alpha)(1 - \frac{2b}{3})$$

$$\geq \frac{2}{3}\sigma'(a)\rho_\varepsilon \rho'_X ||\bar{\beta}|| \frac{b^2}{2}\tan^2(\alpha) .$$

The last inequality uses that $b = \frac{r}{2\sqrt{2}} \leq 1$. Furthermore, for $\alpha \leq \alpha_0 \leq \frac{\pi}{4} : tan(\alpha) \geq \alpha$.

We deduce the curvature inequality:

$$\mathcal{L}(\theta) - \mathcal{L}(\bar{\mu}) \geq \frac{1}{24}\sigma'\left(\frac{||\bar{\beta}||r}{\sqrt{2}}\right)\rho_\varepsilon \rho'_X ||\bar{\beta}|| r^2 \alpha(\theta, \bar{\mu})^2 .$$

$\square$

The next step in proving the bound on $\hat{\alpha}_n$ is to upper bound the excess risk using tools from empirical process theory. In particular, to get fast rates, we will use a localization argument to restrict the suprema of the process $|\mathcal{L}_n(\theta) - \mathcal{L}(\theta)|$ to be over the spherical cap of vectors $\{\theta \in S, \alpha(\theta, \bar{\mu}) \leq \hat{\alpha}_n\}$. For a fixed value of $\hat{\alpha}_n$, we control the suprema in expectation using Dudley's entropy bound and in high probability using Talagrand's inequality to get a high probability fixed point inequality of the form $\hat{\alpha}_n \leq U(\hat{\alpha}_n)$ with U a fixed function. Since the angle $\hat{\alpha}_n$ is itself random, we will need to apply a peeling technique and we first need to show that the high probability bound $\alpha \leq U(\alpha)$ holds uniformly over $\alpha$. Our proof resembles that of Kuchelmeister & Geer (2024).

Let now $\alpha \leq \alpha_0$ be a non random angle value.

Let $\mathcal{F}_s \triangleq \{\theta \in S, \alpha(\theta, \bar{\mu}) \leq s\}$

Let $f_\theta(X, Y) \triangleq (2Y - 1)(\text{sign}(X^\top \theta) - \text{sign}(X^\top \bar{\mu}))$.

We denote $Pf_\theta \triangleq \mathcal{L}(\theta) - \mathcal{L}(\bar{\mu}) = E_{X,Y}(f_\theta(X, Y))$ and $P_n f_\theta = \frac{1}{n}\sum_{i=1}^n f_\theta(X_i, Y_i)$.

Going back to the basic inequality, we have for $\theta \in S$ such that $\alpha(\theta, \bar{\mu}) \leq \alpha_0$:

$$\mathcal{L}(\theta) - \mathcal{L}(\bar{\mu}) = Pf_\theta = Pf_\theta - P_n f_\theta + P_n f_\theta$$
$$\leq \sup_{\mathcal{F}_\alpha} |P_n f - Pf|$$

Notice that we localized the process to be in $\mathcal{F}_\alpha$. This process defines a natural metric towards which it is subgaussian: $d(\theta, \theta')^2 \triangleq ||f_\theta - f'_\theta||^2_{L_2(P)} = E_{X,Y}(||f_\theta(X, Y) - f_{\theta'}(X, Y)||^2)$.

We thus need a fine control of the deviations of the process $\sup_{\mathcal{F}_\alpha} |P_n f - Pf|$ and of its mean. We first state and prove a few useful lemmas. The first lemma states that the excess risk is bounded by a multiple of the angle.

**Lemma 1.** *Suppose that X is a bounded random variable that has density $f_X$ bounded above by C, and let $R > 0$ s.t: $||X|| \leq R$ almost surely. Then for $\theta \in S$:*

$$Pf_\theta \leq L\alpha(\theta, \bar{\mu}) \tag{6}$$

*with $L = 2C\,Vol(B_{d-2}(R))R^2$*

*Proof.* We have

$$Pf_\theta = E_X(|ch(X)|\mathbb{1}(\text{sign}(X^\top\theta) \neq \text{sign}(X^\top\bar\mu))$$
$$\leq \mathbb{P}(\text{sign}(X^\top\theta) \neq \text{sign}(X^\top\bar\mu))$$

Since the density of X is upper bounded by C and X is upper bounded by R, then the density of the projection of X on $V = span(\{\theta, \bar\mu\})$ is upper bounded by $C\,Vol(B_{d-2}(R))$. And we have:

$$Pf_\theta \leq 2C\,Vol(B_{d-2}(R)) \int_{y \in V \cap B_d(R)} \mathbb{1}(y^\top\theta < 0, y^\top\bar\mu > 0) \tag{7}$$
$$\leq 2C\,Vol(B_{d-2}(R))R^2\alpha(\theta, \bar\mu) . \tag{8}$$

$\square$

Note that the constant L does not explode in general with the dimension d since the density itself and its upper bound C typically decrease with d.

We deduce the following corollary that bounds the diameter of $\mathcal{F}_\alpha$ in the $L_2(P)$ metric and the maximum variance of each element of the process.

**Corollary 3.** *We have the following inequalities:*

$$Diam(\mathcal{F}_\alpha, L_2(P)) \leq \sqrt{2L\alpha} \tag{9}$$

*and*

$$\sup_{\theta' \in \mathcal{F}_\alpha} Var_{X,Y}(f_{\theta'}) \leq L\alpha$$

*Proof.* We have for $\theta, \theta' \in \mathcal{F}_\alpha$:

$$d(\theta, \theta')^2 \triangleq ||f_\theta - f'_\theta||^2_{L_2(P)}$$
$$= E_{X,Y}(||f_\theta(X, Y) - f_{\theta'}(X, Y)||^2)$$
$$= E_{X,Y}((2Y - 1)^2(\text{sign}(X^T\theta) - \text{sign}(X^\top\theta'))^2)$$
$$= \mathbb{P}_X(\text{sign}(X^\top\theta) \neq \text{sign}(X^\top\theta'))$$
$$\leq \mathbb{P}_X(\text{sign}(X^\top\theta) \neq \text{sign}(X^\top\bar\mu)) + \mathbb{P}_X(\text{sign}(X^\top\theta') \neq \text{sign}(X^\top\bar\mu))$$
$$\leq 2L\alpha$$

The last inequality being a consequence of the previous lemma. Hence

$$\text{Diam}(\mathcal{F}_\alpha, L_2(P)) \triangleq sup_{\theta, \theta' \in \mathcal{F}_\alpha} d(\theta, \theta') \leq \sqrt{2L\alpha} . \tag{10}$$

We also similarly bound the variance for $\theta \in \mathcal{F}_\alpha$:

$$\text{Var}_{X,Y}(f_\theta) \leq E(f_\theta^2)$$
$$\leq \mathbb{P}_X(\text{sign}(X^\top\theta) \neq \text{sign}(X^\top\bar\mu))$$
$$\leq L\alpha .$$

$\square$

We also state a bound on the metric entropy of $\mathcal{F}_\alpha$ with respect to the distance d which uses that this distance is bounded by the square root of the angle.

**Lemma 2.** *Let $\alpha \leq \alpha_0$ and $\delta > 0$. The covering number of $\mathcal{F}_\alpha$ with respect to d is bounded by:*

$$\log(\mathcal{N}(\mathcal{F}_\alpha, d, \delta)) \leq (d - 1)\log\left(\frac{3\pi\alpha}{2\delta}\right) \tag{11}$$

*Proof.* The proof is standard and upper bounds the packing number to bound the covering one, see for example Wainwright (2019). Let $\alpha \leq \alpha_0$ and $\delta > 0$. Let $S_k$ denote the unit sphere in $R^{k+1}$ and recall that $S \triangleq S_{d-1}$. Let $\sigma$ be the surface measure on $S_k$ (where the dimension is implied by abuse of notation) and $w_k = \sigma(S^k)$ be the surface of the sphere $S_k$. We have by the spherical cap area formula since $\alpha \leq \alpha_0 < \frac{\pi}{2}$:

$$\sigma(\mathcal{F}_\alpha) = w_{d-2} \int_0^\alpha \sin(t)^{d-2} dt$$

Using the inequalities for $0 \leq t \leq \frac{\pi}{2}$:

$$\frac{2}{\pi} t \leq \sin(t) \leq t$$

We have that:

$$w_{d-2}(\frac{2}{\pi})^{d-1} \int_0^\alpha t^{d-2} dt \leq \sigma(\mathcal{F}_\alpha) \leq w_{d-2} \int_0^\alpha t^{d-2} dt \tag{12}$$

$$\frac{w_{d-2}}{d-1}(\frac{2}{\pi})^{d-1} \alpha^{d-1} \leq \sigma(\mathcal{F}_\alpha) \leq \frac{w_{d-2}}{d-1} \alpha^{d-1} \tag{13}$$

where notice in the first line that $(\frac{2}{\pi})^{d-1} \leq (\frac{2}{\pi})^{d-2} \leq 1$. If $\delta \geq \alpha$, then the inequality **??** trivially holds since $\mathcal{F}_\alpha$ can be covered by itself and thus $\mathcal{N}(\mathcal{F}_\alpha, d, \delta) = 1$. Suppose now that $\delta < \alpha$ and let $M := \mathcal{M}(\mathcal{F}_\alpha, d, \delta)$ be the $\delta-$packing number of $\mathcal{F}_\alpha$ which upper bounds the covering number Wainwright (2019). Let $\{x_1, \ldots, x_M\}$ be a maximal packing of size $M$. Then, by maximality $\mathcal{F}_\alpha \subset \cup_{i=1}^K \mathcal{F}_\delta(x_i)$ and $\mathcal{N}(\mathcal{F}_\alpha, d, \delta) \leq K$ where $\mathcal{F}_\delta(x_i) = \{\theta \in S/\alpha(\theta, x_i) \leq \delta\}$. On the other hand, we have for $u \in \mathcal{F}_\delta(x_i)$:

$$\alpha(u, \bar{\mu}) \leq \alpha(u, x_i) + \alpha(x_i, \bar{\mu}) \leq \alpha + \frac{\delta}{2}$$

Hence:

$$\cup_{i=1}^K \mathcal{F}_\delta(x_i) \subset \mathcal{F}_{\alpha+\frac{\delta}{2}}$$

Since $\{\mathcal{F}_\delta(x_i)\}_{i \in [K]}$ are disjoint sets, we deduce using the surface inequalities 13 that:

$$K \frac{w_{d-2}}{d-1}(\frac{2}{\pi})^{d-1} \delta^{d-1} \leq K\sigma(\mathcal{F}_{\frac{\delta}{2}}) \leq \frac{w_{d-2}}{d-1}(\alpha + \frac{\delta}{2})^{d-1}$$

and therefore, since $\delta \leq \alpha$:

$$\mathcal{N}(\mathcal{F}_\alpha, d, \delta) \leq K \leq (\frac{\pi}{2})^{d-1}(1 + \frac{\alpha}{\delta})^{d-1}$$

$$\leq (\frac{3\pi\alpha}{2\delta})^{d-1}$$

which achieves the desired bound.

$\square$

Giving these lemmas, we go back to bounding the supremum of the process $sup_{\mathcal{F}_\alpha}|P_n f - Pf|$. We first bound its expectation using Dudley's entropy integral bound (which is enough for our use case to get the desired bound).

**Theorem 4.** *Let $\alpha \geq 0$. There exists a constant C' that depends on the dimension and L such that:*

$$\mathbb{E}\left[\sup_{\mathcal{F}_\alpha}|P_n f - Pf|\right] \leq C'\sqrt{\frac{(d-1)\alpha}{n}} \tag{14}$$

*Proof.* Note that the process is subgaussian with respect to $d$ and has finite metric entropy. Using Dudley's entropy bound and the previous lemmas, we have for a universal constant

C' (which changes each line with abuse of notation):

$$\mathbb{E}[\sup_{\mathcal{F}_\alpha} |P_n f - Pf|] \leq \frac{C'}{\sqrt{n}} \int_0^{Diam(\mathcal{F}_\alpha, L_2(P))} \sqrt{\log(\mathcal{N}(\mathcal{F}_\alpha, d, \delta))} d\delta$$

$$\leq \frac{C'}{\sqrt{n}} \int_0^{\sqrt{2L\alpha}} \sqrt{(d-1)\log(\frac{3\pi\alpha}{2\delta})} d\delta$$

$$\leq \frac{C'\sqrt{d-1}}{\sqrt{n}} \sqrt{\alpha} \int_0^1 \sqrt{\log(\frac{3\pi\sqrt{\alpha_0}}{4L\delta})} d\delta$$

$$\leq C'\sqrt{\frac{(d-1)\alpha}{n}}$$

using the change of variables formula, $\alpha \leq \alpha_0 \leq \frac{\pi}{4}$ and monotonicity of the logarithm. □

Now, given the bound on expectation and the previous lemmas, we can bound the supremum in high probability using Bousquet's version of Talagrand's inequality. (For simplicity, we omit discussions about measurability, Bousquet's theorem is stated for a countable index family to guarantee the measurability of the supremum but can be extended to our setting as well.)

**Theorem 5.** *There exists a constant $K$ such that with probablity at least $1 - \delta$, we have:*

$$\sup_{\mathcal{F}_\alpha} |P_n f - Pf| \leq K \left[ \sqrt{\frac{\alpha(d - 1 + \log(\frac{1}{\delta}))}{n}} + \frac{\log(\frac{1}{\delta})}{n} \right] \tag{15}$$

*Proof.* Define $Z \triangleq \sup_{\mathcal{F}_\alpha} |P_n f - Pf|$. We have from the previous lemmas : $\sup_{\theta' \in \mathcal{F}_\alpha} \text{Var}_{X,Y}(f_{\theta'}) \leq L\alpha$ and $||f||_\infty \leq 2$. Thus using Bousquet's inequality, with probability at least $1 - \delta$:

$$Z \leq \mathbb{E}(Z) + \sqrt{\frac{2\log(\frac{1}{\delta})(L\alpha + 2\mathbb{E}(Z))}{n}} + \frac{2\log(\frac{1}{\delta})}{3n} .$$

We can simplify the bound using sub-additivity of the square root and (AM-GM) inequality to get:

$$Z \leq 2\mathbb{E}(Z) + \sqrt{\frac{2\log(\frac{1}{\delta})L\alpha}{n}} + \frac{5\log(\frac{1}{\delta})}{3n} .$$

Finally, using the bound derived on $\mathbb{E}(Z)$ and $\sqrt{x} + \sqrt{y} \leq \sqrt{2(x+y)}$, we conclude that:

$$\sup_{\mathcal{F}_\alpha} |P_n f - Pf| \leq K \left[ \sqrt{\frac{\alpha(d - 1 + \log(\frac{1}{\delta}))}{n}} + \frac{\log(\frac{1}{\delta})}{n} \right] .$$

□

Now given the previous concentration inequality and the curvature bound, we can get the desired high probability bound for a fixed $\alpha$ of the form $\alpha \leq U(\alpha)$, what remains is to apply a peeling technique to apply the bound uniformly over $\alpha$ and handle the randomness in $\hat{\alpha}_n$. We restate our finite-sample bound on the rate of convergence of $\hat{\alpha}_n$.

**Theorem.** *Let $\alpha_0 \in [0, \frac{\pi}{4}]$. Let $\delta > 0$. Let $n_0$ be the universal constant such that with probability at least $1 - \delta : \forall n \geq n_0 : \hat{\alpha}_n \leq \alpha_0$. Recall the constant $C_0$ from the curvature inequality and the constant $K$ from the concentration inequality. Then with probability at least $1 - 2\delta, \forall n \geq n_0$:*

$$\hat{\alpha}_n \triangleq \alpha(\hat{\mu}_n^{\text{Sign}}, \bar{\mu}) \leq \left(\frac{K}{C_0}\right)^{\frac{2}{3}} \left(\frac{d - 1 + \log(\frac{\log(n)}{\delta})}{n}\right)^{\frac{1}{3}} + \sqrt{\frac{\log(\frac{\log(n)}{\delta})}{n}}$$

*Proof.* Let $n \geq m_0$. Throughout we condition on the event $\{\hat{\alpha}_n \leq \alpha_0\}$ that has probability greater than $1 - \delta$. We apply a peeling technique for the concentration inequality to hold uniformly. Define $K_n = \lceil \log_2(n) \rceil, \alpha_k = \frac{\alpha_0}{2^k}$ for $k \in [0, K_n]$. Applying the concentration inequality at level $\alpha_k$ with confidence $\frac{\delta}{K_n+1}$ defines the event $\mathcal{E}_k \triangleq \{\sup_{\mathcal{F}_{\alpha_k}} |P_n f - P f| \leq K[\sqrt{\frac{\alpha_k(d-1+\log(\frac{K_n+1}{\delta}))}{n}} + \frac{\log(\frac{K_n+1}{\delta})}{n}]\}$ Thus, by union bound:

$$\mathbb{P}(\Omega \triangleq \cap_{k=0}^{K_n} \mathcal{E}_k) \geq 1 - \sum_{k=0}^{K_n} \frac{\delta}{K_n+1} = 1 - \delta$$

We condition on $\Omega$ for the rest of the proof.

If $\hat{\alpha}_n \leq \frac{\alpha_0}{2^{K_n}} \leq \frac{\alpha_0}{n}$, then the bound is trivial (provided a high enough $n_0$). Otherwise, let k be the unique integer such that $\alpha_{k+1} \leq \hat{\alpha}_n \leq \alpha_k$. Since $\hat{\alpha}_n \in \mathcal{F}_{\alpha_k}$, we have:

$$\mathcal{L}(\hat{\alpha}_n) - \mathcal{L}(\bar{\mu}) \leq \sup_{\mathcal{F}_{\alpha_k}} |P_n f - P f| \leq K \left[ \sqrt{\frac{\alpha_k(d-1+\log(\frac{K_n+1}{\delta}))}{n}} + \frac{\log(\frac{K_n+1}{\delta})}{n} \right] .$$

Since $\hat{\alpha}_n \leq \alpha_0$, we can use the curvature inequality to get the bound on the angle:

$$C_0 \hat{\alpha}_n^2 \leq K \left[ \sqrt{\frac{\alpha_k(d-1+\log(\frac{K_n+1}{\delta}))}{n}} + \frac{\log(\frac{K_n+1}{\delta})}{n} \right]$$

And since $\alpha_k \leq 2\hat{\alpha}_n$, we get the bound:

$$\hat{\alpha}_n^2 \leq \frac{K}{C_0} [\sqrt{\frac{(d-1+\log(\frac{K_n+1}{\delta}))}{n}} \sqrt{\hat{\alpha}_n} + \frac{\log(\frac{K_n+1}{\delta})}{n}] .$$

We can now use the elementary inequality to solve the fixed equation: for $a, b \geq 0$, if $x^4 \leq ax + b$, then $x^2 \leq (2a)^{\frac{2}{3}} + (2b)^{\frac{1}{2}}$. We then get the desired bound on the angle using that $K_n + 1 \leq 2\log(n)$ and by absorbing the numerical constants into K:

$$\hat{\alpha}_n \triangleq \alpha(\hat{\mu}_n^{\text{Sign}}, \bar{\mu}) \leq \left( \frac{K}{C_0} \right)^{\frac{2}{3}} \left( \frac{d-1+\log(\frac{\log(n)}{\delta})}{n} \right)^{\frac{1}{3}} + \sqrt{\frac{\log(\frac{\log(n)}{\delta})}{n}}$$

We now prove Corollary 2 which shows how the non-asymptotic rate behaves as a function of d. Suppose that $X$ follows a uniform distribution in $B(0,1)$ and $\beta \sim \mathcal{N}(\bar{\beta}, \frac{1}{d} I_d)$. We have that:

$$K = \Theta(\log(\frac{A^d}{C_X Vol(B_{d-2}(0,1))})\sqrt{d} + 2\frac{C_X Vol(B_{d-2}(0,1))}{\sqrt{d}})$$

In this case $C_X = \frac{1}{Vol(B_d(0,1))}$ We use the unit ball volume approximation to get:

$$C_X Vol(B_{d-2}(0,1)) = \Theta(\frac{(\frac{2\pi e}{d-2})^{\frac{d-2}{2}}}{(\frac{2\pi e}{d})^{\frac{d}{2}}}) = \Theta(d)$$

And thus by ignoring logarithmic factors in d:

$$K = \Theta(d^{\frac{3}{2}} + \sqrt{d}) = \Theta(d^{\frac{3}{2}})$$

We also have that:

$$C_0 = \Theta(||\bar{\beta}||\sigma'(\frac{||\bar{\beta}||}{\sqrt{2}})\rho_\epsilon \rho_X Vol(B_{d-2}(0,1)))$$

With:

$$\rho_\varepsilon = \inf_{u \in S} \mathbb{P}(|\varepsilon^\top u| \leq \frac{||\bar{\beta}||}{2\sqrt{2}})$$

$$= \mathbb{P}(|\mathcal{N}(0, \frac{1}{d})| \leq \frac{||\bar{\beta}||}{2\sqrt{2}})$$

$$= \Theta(\frac{2}{1 + \exp(-\sqrt{\frac{\pi}{8}} d \frac{||\bar{\beta}||}{2\sqrt{2}})})$$

where the last line is obtained by the logistic approximation of the normal distribution tails. Thus $\rho_\varepsilon = \Theta(1)$. We also have that $\rho_X = C_X = \frac{1}{Vol(B_d(0,1))}$ , and therefore using previous calculations:

$$\rho_X Vol(B_{d-2}(0,1)) = \Theta(\frac{(\frac{2\pi e}{d-2})^{\frac{d-2}{2}}}{(\frac{2\pi e}{d})^{\frac{d}{2}}}) = \Theta(d)$$

Hence we deduce that:

$$C_0 = \Theta(d).$$

Therefore, the rate becomes:

$$\alpha(\hat{\mu}_n^{\text{Sign}}, \bar{\mu}) \lesssim (\frac{K}{C_0})^{\frac{2}{3}}(\frac{d}{n})^{\frac{1}{3}}$$

$$\alpha(\hat{\mu}_n^{\text{Sign}}, \bar{\mu}) \lesssim (\frac{d^2}{n})^{\frac{1}{3}}$$

$\square$

## F    Implementation details

**Selecting a subset of surveys to learn**   We select a subset of 200 personas to learn preferences from from Toubia et al. (2025). To allow different modes of evaluation of heterogeneity, we select this subset in two ways.  The first half of the subset is selected by clustering the users using k-medoids(k $= 100$)on the survey results by summing normalized Hamming distances for categorical results and average per-feature Euclidean distance for the numerical ones. The selected users are the resulting k-medoids. Summary statistics for the demographics section of the surveys from Toubia et al. (2025) are shown in Appendix **??**The second half is randomly selected without replacement among remaining test subjects in the dataset.

**Learning individual preference vectors.**   For each triplet $(z, x_1, x_2)$ in the Helpfulness dataset Bai et al. (2022a). As is common practice, we preprocess the embeddings $\phi(z, x)$ by concatenating the query and output and passing them through gpt-oss-20B. The choice probability is obtained by querying gpt-4o-mini using the person's survey summary. Since we get the exact logits, we can learn individual preference vectors for each user i with a much lower sample complexity relative to using binary signals. Hence we train the preference vector of each user using the cross entropy loss with respect to the true logits: $\mathcal{L}(\theta) = \mathbb{E}_X(\mathbb{P}(Y = 1|X, \beta_i)\log(\sigma(X^\top\theta)) + (1 - \mathbb{P}(Y = 1|X, \beta_i))\log(\sigma(-X^\top\theta)))$. Training is done using Kaiming initialization, an exponential learning rate scheduler with learning rate $10^{-4}$ and $\gamma = 0.97$ and a batch size of 256.  We train for a maximum of 10 epochs with early stopping based on validation loss monitoring and a patience of 2 epochs.

**Learning aggregate preferences**   For the EM and RLHF estimators, we tune over a 20% validation set on their respective losses the learning rate $lr \in \{10^{-6}, 10^{-5}, 10^{-4}, 10^{-3}, 10^{-2}\}$ and batch size $\in \{1, 10, 20, 30, 40, 50\}$ and find optimal values of $lr = 10^{-4}$ and batch size of 50. We also experimented with different optimizers(SGD, Adam,AdamW) and learning rate schedulers (constant, cosine and exponential) and retain SGD and constant learning rate.  For the sign estimator,we train using the approximation $\text{sign}(t) \approx 2\sigma(\lambda t) - 1$ and exponentially anneal $\lambda$ from 1 to a maximum of 15 with an exponential factor of 1.02. We found that values in the low 10s are sufficient to ensure convergence since $\sigma(t) = \frac{1}{1+e^{-t}}$ quickly saturates. We tune the learning rate as well in $\{10^{-6}, 10^{-5}, 10^{-4}, 10^{-3}, 10^{-2}\}$ and retain a learning rate of $lr = 0.01$. We find that such a relatively high learning rate works well since the loss is bounded as opposed to the logistic loss. All experiments are averaged over 20 runs.

## G EM ALGORITHM DETAILS

The EM algorithm alternates between the so-called E-step and M-step until convergence: (E) constructs the expected likelihood—a surrogate of the mixture-likelihood using Bayes rule—and (M) updates the model parameter by optimizing the expected likelihood (Train, 2009, Chap. 14).

---

**Require:** Preference data of n users $\mathcal{D} = \cup_i^n \mathcal{D}_i$; Number of clusters $K$; $\mathcal{H} = \bigcup_{i=1}^K \mathcal{H}_i$; feature map $\phi$ ;Initialization strategy; convergence criterion;

1: Initialize $\{\hat{\beta}_i\}_{i=1}^K$ with initialization strategy.
2: **while** not converged **do**
3:     **for all** $h \in \mathcal{H}$ **do**
4:         **E-step (hard cluster assignment):** assign $h$ to the $i$-th cluster such that

$$i \leftarrow \underset{i \in \{1,\dots,K\}}{\arg\min} \sum_{(\mathbf{z},\mathbf{x}_1,\mathbf{x}_2,y)\in\mathcal{D}_h} \mathcal{L}(\hat{\beta}_{\mathbf{i}}, \mathbf{z}, \mathbf{x}_1, \mathbf{x}_2, \mathbf{y})$$

    where
$\mathcal{L}(\cdot) \quad = \quad y \log\left(\sigma((\phi(z,x_1) - \phi(z,x_2))^\top \hat{\beta}_i)\right) \quad + \quad (1 \quad -$
$y)\log\left(1 - \sigma((\phi(z,x_1) - \phi(z,x_2))^\top \hat{\beta}_i)\right)$ .
5:     **end for**
6:     **M-step:** Update each $\hat{\beta}_i$ for $i = 1,\dots,K$ by minimizing the negative log-likelihood loss $\mathcal{L}$ on the assigned users' data.
7: **end while**
8: **return** $\frac{1}{K}\sum_{i=1}^K \hat{\beta}_i$

---

Algorithm 1: EM algorithm, adapted from Chakraborty et al. (2024)

For the convergence criterion, we run the algorithm until clusters stop updating with a random $\mathcal{N}(\hat{\beta}^{RLHF}, \frac{1}{d}I_d)$ around the mean.

For each $K$, the EM algorithm outputs the mixture components $\hat{\beta}_1^{\text{EM,K}}, \dots, \hat{\beta}_K^{\text{EM,K}}$ and user assignment vector $(\delta_{i,k})_{i \in [n_u], k \in [K]}$; our EM estimate of the population mean utility is $\hat{\beta}^{\text{EM}} = \frac{1}{n_u} \cdot (\sum_{i=1}^{n_i} \sum_{k=1}^K \delta_{i,k} \cdot \hat{\beta}_k^{\text{EM}})$.

## H PROMPT USED TO LABEL

---

**Persona-conditioned annotation prompt**

You are a persona in a survey. You are given a survey that your persona answered and that reflects its preferences. You will be given an extra question at the end, and you will need to answer it based on your persona's preferences. You should answer with just **1 or 2**.

**Persona summary.** The person's demographics are the following...

- **Geographic region**: Northeast (PA, NY, NJ, RI, CT, MA, VT, NH, ME)
- **Gender**: Male
- **Age**: 30-49

$$\vdots$$

The person was finally asked the question "Please describe the type of person you actually are. That is, write about the traits and behaviors you actually possess. Please write at least 3 sentences." They answered: "i'm in a lot of pain that keeps me from doing the things i want to do; even when i have the physical capacity i often can't focus on doing them. i do my best to fulfill my responsibilities, regardless, but i often come short of my own standards. i love to play with words and ideas, i love to create some sort of order out of chaos, i love to learn new things, i love to laugh and make my friends laugh."

---

**Question you will be answering (please answer with just 1 or 2):** If you were the human part in the following conversations with an AI assistant, which option would be more helpful to you? Question type: Single choice .
Options:

1.

**Human:** What are the best movies that are set in space?
**Assistant:** I have a number of recommendations for science fiction movies set in space that might interest you, such as Star Wars, District 9, The Martian, Moon, and Interstellar, as well as classics such as 2001: A Space Odyssey and Star Trek.

2.

**Human:** What are the best movies that are set in space?
**Assistant:** Well it's hard to make a list that works, but here's one we can start with:
"2001: A Space Odyssey"
"Star Wars"
"Blade Runner"
"Galaxy Quest"
"Apollo 13"

## I  LABELLING

| Person 10 Survey Summary | Person 76 Survey Summary |
|---|---|
| The person's demographics are the following... 

 • **Geographic region**: Northeast (PA, NY, NJ, ...) 
 • **Gender**: Male 
 • **Age**: 30-49 
 • **Education level**: Some college, no degree 
 • **Race**: Other 
 • **Citizen of the US**: Yes 
 • **Marital status**: Living with a partner 
 • **Religion**: Other 
 • **Religious attendance**: Once or twice a month 
 The person's Big 5 scores are the following: 
 • **score_extraversion** = 1.75 (15th percentile) 

 $\vdots$ | The person's demographics are the following... 

 • **Geographic region**: South (TX, OK, AR, LA, KY, TN, MS, AL, WV, DC, MD, ...) 
 • **Gender**: Female 
 • **Age**: 65+ 
 • **Education level**: Some college, no degree 
 • **Race**: White 
 • **Citizen of the US**: Yes 
 • **Marital status**: Widowed 
 • **Religion**: Other 
 • **Religious attendance**: More than once a week 
 The person's Big 5 scores are the following: 
 • **score_extraversion** = 3.375 (71st percentile) 

 $\vdots$ |

| RLHF Alternative 1 | RLHF Alternative 2 |
|---|---|
| **Human:** What should I do if my boss is always mean to me? 
 **Assistant:** I'm sorry to hear about that. Do you want to tell me about the situation, or would you like for me to make some suggestions? 
 **Human:** He is ignorant and mean. I feel bad going to work. What should I do? 
 **Assistant:** I'm sorry you feel that way. Did you notice if he is nice to other people? Or does he treat you in a way that's different from other people? 
 **Human:** He is mean to about half of us. I don't know what his problem is. 
 **Assistant:** Would you like to tell me about what specifically you find bad about his behavior? Would it help to explain why you're worried about his behavior and what you think is the worst about it? And I'm curious, why do you think he's mean? What are some of the things he does that make you feel this way? | **Human:** What should I do if my boss is always mean to me? 
 **Assistant:** I'm sorry to hear about that. Do you want to tell me about the situation, or would you like for me to make some suggestions? 
 **Human:** He is ignorant and mean. I feel bad going to work. What should I do? 
 **Assistant:** I'm sorry you feel that way. Did you notice if he is nice to other people? Or does he treat you in a way that's different from other people? 
 **Human:** He is mean to about half of us. I don't know what his problem is. 
 **Assistant:** He doesn't treat you like the others. Maybe he's unfair about something. |

| Person 10 Choice Logits | Person 76 Choice Logits |
|---|---|
| token "1" : -0.1602 
 token "2" : -1.9102 
 token "3" : -18.285 
 token "**" : -19.785 

 $\vdots$ | token "1" : -0.6931 
 token "2" : -0.6931 
 token "3" : -17.318 
 token "I" : -19.193 

 $\vdots$ |

| Renormalized choice probability | Renormalized Choice Probability |
|---|---|
| $\mathbb{P}[\text{Choose alternative 1 over 2}] = 0.851$ | $\mathbb{P}[\text{Choose alternative 1 over 2}] = 0.5$ |

Figure 5: **Representative persona contrasts and RLHF labelling simulation.** Top: concise survey summaries. Center: Example of two RLHF alternatives considered. Bottom: raw logits and renormalized choice probabilities.

## J  EXAMPLE OF PERSONA SUMMARY

We show an example of a persona's summary (persona 1) from Toubia et al. (2025)

**Persona 1 Survey Summary**

The person's demographics are the following...
Geographic region: Midwest (ND, SD, NE, KS, MN, IA, MO, WI, IL, MI, IN, OH)
Gender: Female
Age: 30-49
Education level: Some college, no degree
Race: Black
Citizen of the US: Yes
Marital status: Married
Religion: Nothing in particular
Religious attendance: A few times a year
Political affiliation: Independent
Income: $50,000-$75,000
Political views: Moderate
Household size: 3
Employment status: Part-time employment

The person's Big 5 scores are the following:
score_extraversion = 2.125 (26th percentile)
score_agreeableness = 4.556 (84th percentile)
wave1_score_conscientiousness = 4.556 (77th percentile)
score_openness = 3.5 (36th percentile)
score_neuroticism = 1.5 (15th percentile)

Openness reflects curiosity and receptiveness to new experiences, Conscientiousness indicates self-discipline and goal-directed behavior, Extraversion measures sociability and assertiveness, Agreeableness reflects compassion and cooperativeness, and Neuroticism captures emotional instability and susceptibility to negative emotions. Each score ranges from 1 to 5, and a higher score indicates a greater display of the associated traits. The person's need for cognition score is the following: score_needforcognition = 2.611 (18th percentile)
Need for cognition is a personality trait that reflects an individual's tendency to seek out, engage in, and enjoy complex cognitive tasks. The score ranges from 1 to 5, and a higher score indicates a higher need for cognition.

The person's agentic / communal value scores are the following:
score_agency = 7.583 (95th percentile)
score_communion = 7.583 (71st percentile)
Agency is the meta-concept associated with self-advancement in social hierarchies; communion is the partner concept associated with maintenance of positive relationships. Each score ranges from 1 to 9, and higher values indicate higher propensity for these constructs.

The person's minimalism score is the following:
score_minimalism = 4.417 (91st percentile)
The score ranges from 1 to 5, and a higher score indicates a higher preference for minimalism.

The person's basic empathy scale score is the following: score_BES = 3.7 (38th percentile)
The score ranges from 1 to 5, and a higher score indicates more empathy.

The person's G.R.E.E.N. score is the following:
score_GREEN = 4 (68th percentile)
The score ranges from 1 to 5, and higher scores indicate a higher affinity for environmentalism.

The person's CRT score is the following:
crt2_score = 2 (59th percentile)
The score ranges from 0 to 4, and a higher score indicates a greater ability to suppress an intuitive and spontaneous ("system 1") wrong answer in favor of a reflective and deliberative ("system 2") right answer.

The person's fluid and crystallized intelligence scores are the following:
score_fluid = 1 (55th percentile)
score_crystallized = 1 (4th percentile)
Fluid intelligence is the capacity to reason, solve novel problems, and adapt to new situations independent of prior knowledge, while crystallized intelligence is the accumulation of knowledge, facts, and skills acquired through experience and education. The fluid score ranges from 0 to 6, and the crystallized score ranges from 0 to 20; higher scores indicate better performance.

The person's syllogism score is the following:
score_syllogism_merged = 5 (32nd percentile) The score ranges from 0 to 12, and a higher score indicates a greater ability to solve verbal reasoning problems like "All A are B" and "All B are C" implying "All A are C."

The person's total intelligence scores, overconfidence score, and overplacement score are the following:
score_actual_total = 9 (8th percentile)
score_overconfidence = 1 (8th percentile)
score_overplacement = -15 (3rd percentile)
The total intelligence score is simply the sum of the person's performances on the aforementioned logic / intelligence questions (ranging from 0 to 42 total correct). The person's overconfidence is the difference between their prediction of their own performance and their actual performance. The person's overplacement is the difference between their prediction of their own performance and their prediction of other respondents' performance.

The person's ultimatum game scores are the following:
score_ultimatum_sender = 0 (6th percentile)
score_ultimatum_accepted = 100 (100th percentile)
The sender score is the percent of $5 the person chose to offer another person in the ultimatum game. The receiver score is the percent of offers they accepted, out of a total of 6 offers made in $1 increments from $5 to $0, when acting as the receiver in the game.

The person's mental accounting score is the following:
score_mentalaccounting = 25 (9th percentile)
The score ranges from 0 to 100 percent, and higher scores indicate a greater adherence to the principles of mental accounting proposed by Thaler: segregate gains, integrate losses, segregate a small gain from a large loss, and integrate a small loss with a large gain.

The person's social desirability score is the following:
score_socialdesirability = 5 (48th percentile)
The score ranges from 0 to 13, and higher scores indicate a greater tendency to respond to questions in a socially desirable way rather than in a truthful way.

The person's secondary conscientiousness score is the following:
wave2_score_conscientiousness = 8 (100th percentile)
This score was computed using a different questionnaire than the Big 5 conscientiousness score reported above, and it ranges from 0 to 8, but it is otherwise similar. Higher

scores indicate a greater propensity for conscientiousness.

The person's Beck anxiety score is the following:
score_anxiety = 27 (93rd percentile)
The score ranges from 0 to 63, and higher scores indicate a higher tendency for anxiety.

The person's individualism vs collectivism scores are the following:
score_HI = 4.25 (52nd percentile)
score_HC = 3.75 (45th percentile)
score_VI = 4.5 (98th percentile)
score_VC = 5 (100th percentile)
The Horizontal Individualism (HI) score reflects a person's preference for autonomy and equality, the Horizontal Collectivism (HC) score captures a preference for interdependence and equality, the Vertical Individualism (VI) score indicates a drive for personal achievement and acceptance of hierarchical inequality, and the Vertical Collectivism (VC) score denotes a focus on group loyalty combined with acceptance of hierarchical structures. Each score ranges from 1 to 5, and a higher score indicates a greater preference.

The person's financial literacy score is the following:
score_finliteracy = 3 (15th percentile)
The score ranges from 0 to 8, and a higher score indicates the person correctly answered more questions related to general financial literacy.

The person's numeracy score is the following:
score_numeracy = 3 (20th percentile)
The score ranges from 0 to 8, and a higher score indicates the person correctly answered more questions related to numeracy.

The person's modus ponens deductive certainty score is the following:
score_deductive_certainty = 4 (100th percentile)
The score ranges from 0 to 4, and a higher score indicates the person correctly answered more modus ponens questions.

The person's forward flow score is the following:
score_forwardflow = 0.801 (31st percentile)
A higher score indicates the person was able to generate more distant words in a sequence (e.g. "candle -> bee -> sugar" instead of "candle -> fire -> flame"). The actual word chain that the subject generated was the following: paper -> pencil -> eraser -> marker -> pen -> point -> sharp -> edge -> cliff -> mountain -> sky -> blue -> cloud -> soft -> pillow -> sock -> pair -> black -> wing -> flap

The person's discount rate and present bias are the following:
score_discount = 0 (40th percentile)
score_presentbias = 0 (52nd percentile)
These are implied rates computed from the person's time-value of money preferences. Higher values of the discount rate imply greater impatience. Higher values of present bias imply greater departure from normative economic behavior.

The person's risk aversion score is the following:
score_riskaversion = 0.306 (79th percentile)
Higher scores indicate a greater tendency for risk aversion in a choice between a sure-amount and lottery payout.

The person's trust game scores are the following:
score_trustgame_sender = 40 (61st percentile)
score_trustgame_receiver = 33 (29th percentile)
The sender score is the percent sent in the trust game (where one player sends money to another, the money is multiplied, and the second player can return some to the first player). The person was asked to list up to 6 thoughts that crossed their mind while playing as the sender and they answered: "half; fairness; not greedy; decent amount; undecided; back way". The person was then asked to list up to 6 thoughts that crossed their mind while playing as the receiver and they answered: "thoughtful; not greedy; good price; split it; wanting more"

The person's regulatory focus scale score is the following:
score_RFS = 5.6 (87th percentile)
The score ranges from 1 to 7. Higher scores indicate a stronger orientation toward either promotion or prevention focus, meaning individuals would be more driven by promotional aspirations or more motivated by avoiding losses.

The person's tightwad-spendthrift score is the following:
score_ST-TW = 7 (10th percentile)
The score ranges from 4 to 26. Lower scores (4-11) indicate difficulty spending money, while higher scores (19-26) indicate difficulty controlling spending.

The person's Beck depression score is the following:
score_depression = 12 (64th percentile)
The score ranges from 0 to 61, and a higher score indicates more depressive behaviors.

The person's need for uniqueness score is the following:
score_CNFU-S = 2.583 (57th percentile)
The score ranges from 1 to 5, and higher scores indicate a higher need for uniqueness.

The person's self-monitoring score is the following:
score_selfmonitor = 3.615 (98th percentile)
The score ranges from 0 to 5, and higher scores indicate a higher ability to monitor one's own behavior.

The person's self-concept clarity score is the following:
score_SCC = 3.417 (42nd percentile)
The score ranges from 1 to 5, and higher scores indicate a greater certainty about the person's own self-concept.

The person's need for closure score is the following:
score_needforclosure = 3.867 (71st percentile)
The score ranges from 1 to 5, and higher scores indicate a greater desire for certainty over ambiguity.

The person's maximization scale score is the following:
score_maximization = 4.667 (99th percentile)
The score ranges from 1 to 5, and higher scores indicate a tendency to optimize rather than satisfice when making decisions.

The person's Wason selection score is the following:
score_wason = 3 (96th percentile)
The score ranges from 0 to 4, and higher scores indicate better performance on the Wason selection task.

The person's dictator game score is the following:

score_dictator_sender = 20 (18th percentile)

This is the percent split sent when the person played the dictator game as the sender. The person was asked to list up to 6 thoughts that crossed their mind when playing this game and they answered: "give money; generous; control; thoughtful; kind; not wasteful"

The person also answered three purely qualitative questions about their concept of self. The person was asked, "Please describe the type of person you aspire to be. That is, write about the traits and behaviors you would like ideally to possess, your ultimate goals for yourself. Please write at least 3 sentences."

They answered: "I would like to be a less selfless person that is in thought someone who has a little more patience and self control. My tendency to be selfish is the most important thing for me because even though I would do anything for my family my initial thoughts are to not do anything at first in my brain."

The person was then asked the question "Please describe the type of person you ought to be. That is, write about the traits and behaviors attributes that you should or ought to possess, based on your responsibilities and what other people expect from you. Please write at least 3 sentences."

They answered: "I ought to be more helpful. Sometimes I'm just lazy because I have so much to do and then I put it off because I'm so overwhelmed. So when I need to help with dishes every night, sometimes I put it off, and then my husband who works 12 hours a day and does them. "

The person was finally asked the question "Please describe the type of person you actually are. That is, write about the traits and behaviors you actually possess. Please write at least 3 sentences."

They answered: "I am so generous. I would put everyone's needs before mine every time. I am a loving , funny person who always has a joke to tell and laugh at myself all the time."

