# OpenReview forum: "The Sign Estimator: LLM Alignment in the Face of Choice Heterogeneity"
_ICLR.cc/2026/Conference — Submitted to ICLR 2026_

### Official Review · Reviewer_vJr6 · 2025-10-27

**Soundness:** 3
**Presentation:** 2
**Contribution:** 3
**Rating:** 6
**Confidence:** 4

**Summary:**

This paper studies reward-model learning for RLHF under heterogeneous human preferences. It shows that the current existing reward model via cross-entropy/MLE is mis-specified. For linear models, the existing RLHF estimator recovers a variance-weighted average of individual utilities, which overweights uncertain users and underweights confident ones rather than the population mean utility direction. The authors propose the Sign estimator, which replaces cross-entropy with a 0–1 sign-agreement objective on pairwise comparisons. Under a mild symmetry assumption on heterogeneity, the Sign estimator recovers the ordinal preferences of the population mean and the direction of the utility vector in the linear case. They further prove a finite-sample convergence rate of $\tilde{\mathcal{O}}(n^{-1/3})$ for the angular error in the linear setting under mild conditions. Empirically, it beats a simple EM-based panel method while keeping the standard pipeline.

**Strengths:**

1. The mis-specifiation of the existing RLHF estimator is an interesting finding of this paper. Proposition 1 shows that MLE under heterogeneity amplifies the influence of the user's responses with high variances and ignores the users who are decisive. This is a valuable conceptual contribution.
2. The Sign estimator is simple and consistent, with rigorous theoretical support in section 4. it guarantees ordinal consistency under mild symmetry and achieves a provable $\tilde{\mathcal{O}}(n^{-1/3})$ convergence rate.
3. The empirical results in section 5 are compelling compared to the RLHF estimator. On a controlled heterogeneous-preference benchmark, the Sign estimator consistently outperforms both the standard RLHF baseline and an EM mixture model, demonstrating its practical advantage without complicating the training pipeline.

**Weaknesses:**

1. My main concern is that the sample complexity analysis of the sign estimator misses the realistic regime of the dimensionality versus sample size. In practice, $d$ could be much larger than $n$, yet here in Theorem 2, it is assumed that the $n\ge n_0$ so that the angular error is small with high probability. It seems the regime of small $n$ is missing. Also, $K$ and $C_{0}$ in Theorem 2 depend on $d$ as well, yet the paper mainly treats $d$ as a constant. Specifically, the result $\tilde{\mathcal{O}}(n^{-1/3})$ ignores $d$ as well. The results can be strengthened if the sample complexity for small $n$ is also presented, which might be more insightful for practical scenarios.
2. In the presentation aspect, there are some terms or phrases lack of explanation. For example, in line 150, I am not sure what "dense" means.

---
Minors:
1. equation in line 262-263 should be for $u'$ instead of $\hat{u}^{\text{Sign}}$.
2. there is an extra "reduce" in line 384-385.

**Questions:**

1. It would be better if there is a slightly more detailed review of the existing results in the heterogeneous preference settings for the sample complexity analysis. The rate $\Theta(n^{-O(1/d)})$ is obtained under what conditions? Are they the same as the settings considered in this paper?
2. When interpret the results of Theorem 2, can we also get a rate treating $d$ as a variable as well? It seems to me that the rate is roughly $O(d^{4/3} n^{-1/3})$ if ignoring $C_0$.
3. The angular results in Section 5 show that the sign estimator has smaller error. Yet, both estimators have pretty large angular errors (all greater than 40 degree even n is ~$10^5$). Could you explain why is this? Could this result be regarded a rigorous support for the competiveness of the proposed estimator?

---

> ### Author Response · Authors · 2025-11-22
> **Response to Reviewer vJr6 [Part 1/2]**
>
> We are thankful for your time and help, especially with regard to dimensionality analysis. We were pleased to see you highlight our analysis of how the standard RLHF procedure amplifies the influence of the user’s responses with high variance while ignoring the users who are decisive. We believe that this only makes the need to address RLHF under heterogeneity more pressing, as ignoring it may lead to undesirable model behavior.
>
> > My main concern is that the sample complexity analysis of the sign estimator misses the realistic regime of the dimensionality versus sample size. In practice, $d$ could be much larger than $n$, yet here in Theorem 2, it is assumed that the $n\ge n_0$ so that the angular error is small with high probability. It seems the regime of small $n$ is missing. Also, $K$ and $C_{0}$ in Theorem 2 depend on $d$ as well, yet the paper mainly treats $d$ as a constant. Specifically, the result $\tilde{\mathcal{O}}(n^{-1/3})$ ignores $d$ as well. The results can be strengthened if the sample complexity for small $n$ is also presented, which might be more insightful for practical scenarios.
>
>  This is a very interesting observation. Indeed, d is in practice comparable to n and our non-asymptotic rates account for that.
>  The threshold  $n_0$ in Theorem 2 is just the minimal sample size n such that the angle is smaller than 90 degrees (the proof for bounding the angle requires the inequality $sin(\alpha) \geq \frac{2}{\pi} \alpha$ to hold, which occurs for small n, even empirically).
>
> However, it is very instructive to determine how our sample error bound varies with the dimension d as you pointed out. In Corollary 2, we instantiate the distributions of $X$ and $\beta$ and derive the exact dependence in $d$. For simplicity, we assume that X is uniform in the unit ball and $\beta \sim \mathcal N(0, \frac{1}{d} I_d)$ so that $\mathbb E(|| \beta||^2)$ is normalized to 1 (the same rate holds under the relaxed assumption that the density of X is bounded away from 0). As you rightfully pointed, the parameter K behaves like $d^{3/2}$. We show that $C_0 $ itself scales as $d^{2/3}$, resulting in an upper bound on $\alpha(\mu_{sign},\bar{\mu}) $ in the order of $(\frac{d^2}{n})^{1/3}$.
>
> We can contrast this rate with the best-known results in the absence of heterogeneity [1] . Here, the BTL model is recovered with rate $||\theta_{BTL} - \theta^\ast\rVert_{\Sigma} \leq \sqrt{\frac{d}{n}}$. Crucially, this rate corresponds to an $O(\sqrt{d^2/n})$ upper bound in an apple-to-apple comparison with our paper. Indeed, the norm $||\theta_{BTL} - \theta^\ast\rVert_{\Sigma}$ in [1] is with respect to the feature covariance $\Sigma = \mathbb E((\phi(X_1) - \phi(X_2)) (\phi(X_1) - \phi(X_2))^T)$ which behaves roughly like $\frac{1}{d} I_d$. It follows that $||\theta - \theta^\ast\rVert_{\Sigma} = \sqrt{(\theta - \theta^\ast) ^T \Sigma (\theta - \theta^\ast)} = \frac{1}{\sqrt d} ||\theta - \theta^\ast||$ and thus $||\theta - \theta^*||_2 \leq \sqrt{d^2 / n}$. Rates scaling as $\frac{d^2}{n}$ are prevalent in recent literature on the regret analysis of RLHF even under the homogeneity assumption [2].
>
> > In the presentation aspect, there are some terms or phrases lack of explanation. For example, in line 150, I am not sure what "dense" means.
>
> Fixed: Apologies for the confusion; we just require that the Lebesgue measure of the set $\Chi$ to be non-zero..
>
> > Minors: equation in line 262-263 should be for  u’ instead of u_sign
>
> Fixed!
> > there is an extra "reduce" in line 384-385.
>
> Fixed.
>
> > Questions: It would be better if there is a slightly more detailed review of the existing results in the heterogeneous preference settings for the sample complexity analysis. The rate $O(n^{1/d})$  is obtained under what conditions? Are they the same as the settings considered in this paper?
>
> Thank you for allowing us to clarify.
>
> While there is long-standing literature on mixed logits, identifiability has been established relatively recently by [3]. This work focuses on identifiability (rather than finite-sample recovery rates) and derives a sufficient moment condition that generalizes Assumption 2. While this work does not tackle efficient estimation, the nonparametric estimator that recovers the full mixing distribution in this setting will typically suffer from the usual curse of dimensionality, with polynomial rates whose exponent shrinks with d, contrary to Theorem 2. [4] propose a nonparametric estimator using deconvolution techniques. This approach achieves L2 convergence rates of order $O(n^{-\gamma s})$ where $\gamma <= \min\{1/(2s + 2d -1),1/(d-1)\}$, assuming that the density of the mixing distribution has Sobolev smoothness s (which our work does not require). In a setting comparable to our Assumptions 1 and 2, this essentially gives a bound whose exponent degrades linearly in $d$.  Note, however, that this bound holds without our symmetry assumption.

---

> > ### Author Response · Authors · 2025-11-22
> > **Response to Reviewer vJr6 [Part 2/2]**
> >
> > It is fair to say that these estimators in this literature are also more complex as they aim to recover the mixture distribution. Targeted estimators for the mean utility can exploit identifiability from Assumptions 1 and 2 to get a consistent estimator with the same exponential dependence in d. The idea is intuitive: for close enough $X_1$ and $X_2$,  the choice probability of a random individual choosing 1 over 2  is $\mathbb E_\beta( \sigma((\phi(X_1) - \phi(X_2) ^T \beta))) \approx ½ + ¼ (\phi(X_1) - \phi(X_2) ^T \mathbb E(\beta)))$. An estimator can thus be obtained by restricting attention to samples $X_1,X_2$ within a ball of radius  \sigma_n that decreases with n. The fraction of samples in such a ball degrades exponentially with d, which yields the desired rate.
> > There is also a literature on latent-class mixture models that exploits connections to tensor/matrix completion but requires linear independence between the utility vectors of the mixture components.
> >
> > > When interpret the results of Theorem 2, can we also get a rate treating  as a variable as well? It seems to me that the rate is roughly  if ignoring .
> >
> > Thank you for the important question. Addressed in weakness 2.
> >
> > > The angular results in Section 5 show that the sign estimator has smaller error. Yet, both estimators have pretty large angular errors (all greater than 40 degree even n is ~). Could you explain why is this? Could this result be regarded a rigorous support for the competiveness of the proposed estimator?
> >
> > This is a great point, the experiment setting is exactly in the case where d is comparable to n and we have $d \leq n \leq d^2$. The experiment is run using GPT-OSS 20B which has a hidden dimension of $\sim 3000$. The angular error is expected to be large in this setting when viewed with respect to the L2 norm, as this norm incurs an extra d factor compared to the $\Sigma$-norm as discussed in weakness 1. The accuracy plots themselves show that we can already recover a 92% accuracy of the choices made by the ground truth $\bar{\beta}$ in this high-dimensional setting.
> >
> > [1]  Banghua Zhu, Jiantao Jiao, and M.I. Jordan. 2023. Principled reinforcement learning with human feedback from pairwise or K-wise comparisons. arXiv:2301.11270.
> >
> > [2] Tengyang Xie, Dylan J Foster, Akshay Krishnamurthy, Corby Rosset, Ahmed Awadallah, and Alexander Rakhlin. Exploratory preference optimization: Harnessing implicit q*-approximation for sample-efficient rlhf. arXiv preprint arXiv:2405.21046, 2024.
> >
> > [3] Fox, J. T., K. I. Kim, S. P. Ryan, and P. Bajari, “The Random Coefficients Logit Model Is Identified,” Journal of Econometrics 166 (2012), 204–12.
> >
> > [4] Gautier, E., Y. Kitamura 2013: Nonparametric Estimation in Random Coefficients Binary Choice Models, Econometrica, 81 (2), 581--607.

---

> > > ### Comment · Reviewer_vJr6 · 2025-11-25
> > >
> > > I appreciate the authors’ thorough and detailed responses. Having considered the clarifications and the overall quality of the paper, I will maintain my current score.

---

### Official Review · Reviewer_wnm8 · 2025-10-31

**Soundness:** 2
**Presentation:** 1
**Contribution:** 2
**Rating:** 2
**Confidence:** 3

**Summary:**

This paper aims to address the preference heterogeneity problem in LLM alignment. Traditional RLHF method treats pairwise data as a single average utility function and fit models with cross entropy. This paper finds the distorted utility. By replacing cross-entropy with binary classification loss, this paper is able to reach polynomial error bounds. Besides, simulations of LLM alignment using digital twins shows improvement over baselines.

**Strengths:**

- The analysis of estimators' bias look interesting and original.
- The empirical evaluation result looks good.

**Weaknesses:**

- The writing needs to be significantly improved.

    - Improper references. For example,  Harsanyi's theorem is reference in the second footnote but not in the first foot note. Similarly, the BTL model is not reference in the second paragraph of the first page (or between 129, 130).
    - $\mathcal{X}$ is said to represent "all (prompt, completion) pairs" in line 119 whereas line 139 samples are drawn from $\mathcal{X}^2$ which implies the pair can have different prompt which is not the common case.
    - For example, line 125 mentions that $\{ \xi_{x,\beta} \}$ is drawn i.i.d. from same distribution. Does it mean we can simply drop the subscript?
    - The $\mu$ is used to denote a distribution in line 137 while $\hat{\mu}$ is used as an estimate of utility function in line 161.
    - Line 161 use RLHF as the superscript but it does not have the RL part.
    - Line 272, dangling ).
    - Too many "striking" in the paper.
    - Too many "some" in the paper. Eg. "some set" in line 37, "some distribution" in line 137.
    - ...

- The rates need to compared with related works such as (zhu et al., 2023).

While I find the topic to be interesting, the current paper is clearly not polished enough for readers and certainly not ready for publication.

**Questions:**

See weakness.

---

> ### Author Response · Authors · 2025-11-22
> **Response to Reviewer wnm8**
>
> > Improper references. For example, Harsanyi's theorem is reference in the second footnote but not in the first foot note. Similarly, the BTL model is not reference in the second paragraph of the first page (or between 129, 130).
>
> Fixed, we have added references. It is incorrect to associate the model described in the second paragraph of the first page to the BTL; it describes a more general random utility choice model.
>
> >  $\mathcal{X}$ is said to represent "all (prompt, completion) pairs" in line 119 whereas line 139 samples are drawn from $\mathcal{X}^2$ which implies the pair can have different prompt which is not the common case.
>
> Fixed in line 151.
>
> > For example, line 125 mentions that $\zeta_{x,\beta}$is drawn i.i.d. from same distribution. Does it mean we can simply drop the subscript?
>
> The convention is to have different indices for IID draws.
>
> > The $\mu$ is used to denote a distribution in line 137 while $\hat{\mu}$ is used as an estimate of utility function in line 161.
>
> Fixed!
> > Line 161 use RLHF as the superscript but it does not have the RL part.
>
> We refer to it as such since it is the standard reward learning procedure used in RLHF.
> > Line 272, dangling ).
>
> Fixed.
> > Too many "striking" in the paper.
>
> Fixed.
> > Too many "some" in the paper. Eg. "some set" in line 37, "some distribution" in line 137.
>
> Fixed.
>
> > The rates need to compared with related works such as (zhu et al., 2023).
>
> We clarify the relationship with [1]. [1] studies the rate of convergence of the standard RLHF model without latent heterogeneity; here, all data comes from a single ground truth reward model. In this context, Maximum Likelihood Estimation is well-specified and tractable, which enables efficient $\sqrt{\frac{d^2}{n}}$ rates. However, such rates don’t apply in the more complex heterogeneous setting that our work considers. Previous work [2] [3] has shown that the estimator in [1] is not even consistent under latent heterogeneity. Our work determines precisely to which solution the estimator in [1] converges. Our main contribution is to design a simple, consistent estimator in the heterogeneous setting that achieves a rate of $(\frac{d^2}{n})^{\frac{1}{3}}$.
> To put this result into perspective, finite-sample recovery results for mixture logit models are scant in previous literature and typically require stringent assumptions on the mixture components. We kindly refer the reviewer to our discussion on Page 8 of the revised paper.
>
> > While I find the topic to be interesting, the current paper is clearly not polished enough for readers and certainly not ready for publication.
>
> We are sorry to hear that you found the paper not polished enough. We have made targeted efforts to improve the writing and fix any other typos. Thank you for your comments above. Please let us know if there is anything else to address.
>
> [1] Banghua Zhu, Jiantao Jiao, and M.I. Jordan. 2023. Principled reinforcement learning with human feedback from pairwise or K-wise comparisons. arXiv:2301.11270.
>
> [2] Anand Siththaranjan, Cassidy Laidlaw, and Dylan Hadfield-Menell. Distributional preference learning: Understanding and accounting for hidden context in RLHF. arXiv preprint arXiv:2312.08358, 2023
>
> [3] Paul Gölz, Nika Haghtalab, and Kunhe Yang. Distortion of ai alignment: Does preference optimization optimize for preferences? arXiv preprint arXiv:2505.23749, 2025

---

### Official Review · Reviewer_rvPb · 2025-11-01

**Soundness:** 2
**Presentation:** 3
**Contribution:** 2
**Rating:** 4
**Confidence:** 4

**Summary:**

- The paper shows that standard reward modeling (using cross-entropy loss) introduces bias by "re-weighting" user preferences. It "amplifies the influence of uncertain users" while "diminishing that of confident ones," effectively ignoring users who have the strongest preferences.
- The authors introduce the "sign estimator," a new method that is a simple drop-in replacement for the cross-entropy loss function , instead using a binary classification (0-1) loss to maximize the agreement between the sign of the estimated utility and the observed preference choice.
- The sign estimator is proven to have some nice theoretical properties, e.g. having "ordinally consistent" preferences (i.e., the correct ranking of choices) under a mild assumption of symmetric preference heterogeneity.
- Some experiments are run to empirically verify that it works better than standard reward modeling.

**Strengths:**

The paper's strengths are on the theoretical end:
- The proposed "sign estimator" is a "drop-in replacement for cross-entropy loss", making it practical to deploy as it "maintain[s] the implementation simplicity of existing LLM alignment pipelines.
- The sign estimator provides a simple, provably consistent, and efficient estimator that "recovers consistent ordinal alignment under mild assumptions".
- The estimator has strong finite-sample guarantees: It achieves the first "polynomial finite-sample error bounds in this setting".

**Weaknesses:**

The paper has several weaknesses on the empirical end:
- The motivation seems to be heterogeneity of human preferences, yet the authors do not use real human preferences, despite there being real human preference datasets that can be used to align LLMs (e.g., SHP or StackExchange). This is important because real human preferences have noise and bias that may render the proposed estimator no better than the standard one. Simulated preferences are simply not enough.
- The chosen evaluation metrics are bizarre: the estimators are evaluated on how well they recover simulated preferences and on an embedding-based metric. Why not actually put the reward model in a loop with an LLM and post-train it? Then you run many of the standard evals (e.g., Alpacaeval2 for instruction-following). As it stands, there is no real evidence here that the proposed estimator will be practically useful for aligning models with preferences.

My recommendation would be to either make this a pure theory paper and submit it AISTATS (or a similar venue) or evince the practical benefit of the sign estimator by showing that it works on real human preferences on LLMs of non-trivial size.

**Questions:**

See the weaknesses section above.

---

> ### Author Response · Authors · 2025-11-22
> **Response to Reviewer rvPb [Part 1/2]**
>
> Thank you for your constructive feedback. Based on your comments, we have expanded our numerical study to make the practical relevance, which motivates our work, more explicit.
>
> > The motivation seems to be heterogeneity of human preferences, yet the authors do not use real human preferences, despite there being real human preference datasets that can be used to align LLMs (e.g., SHP or StackExchange). This is important because real human preferences have noise and bias that may render the proposed estimator no better than the standard one. Simulated preferences are simply not enough.
>
>
> Thank you for allowing us to clarify and expand our numerical study.
>
> We can indeed use real human preference datasets (such as SHP or Harmfulness/Helpfulness data) to compare the performance of preference model estimators out-of-sample. We do just this in the revised paper. This type of experiment shows significant gaps between our method and standard RLHF. An important takeaway from these experiments was that the EM-based baselines were even more unstable in real-world data and not guaranteed to converge, as they are known to be brittle and to suffer from posterior collapse [1], as opposed to algorithms that directly minimize a loss function such as RLHF and our sign estimator. Thank you for the suggestion. It facilitates the comparison with SOTA in existing literature.
> We would like, however, to explain the limitations of such an approach. First, it can evaluate accuracy/disagreement on hold-out data, but it cannot explicitly measure alignment under heterogeneity (i.e., ability to recover the expected reward model up to scaling) because the ground truth is unknown. Second, existing preference data may not reflect real-world heterogeneity in the LLM user population. Recognizing these challenges, [4] [5] proposed a set-up with up to four user segments. Our methodology, which uses LLM digital twins calibrated to a representative panel of 200 real-world individuals, expands on the literature and provides a rigorous empirical set-up to capture the impact of heterogeneity. In hindsight, this may have been unclear in our original submission. This numerical set-up may itself be a contribution to the literature by providing a methodology to evaluate preference models under realistic levels of heterogeneity with ground truth access.
>
> > The chosen evaluation metrics are bizarre: the estimators are evaluated on how well they recover simulated preferences and on an embedding-based metric. Why not actually put the reward model in a loop with an LLM and post-train it? Then you run many of the standard evals (e.g., Alpacaeval2 for instruction-following). As it stands, there is no real evidence here that the proposed estimator will be practically useful for aligning models with preferences.
>
> Our metrics are extensively used in prior work on preference modeling (PM) [2] [4] [5] [6]. Disagreement is 1-Accuracy; we make this more explicit in the revised paper.
>
> We acknowledge that evaluating the benefit of improved preference modeling on downstream LLM alignment (by evaluating a post-trained LLM) is an important criterion. However, existing literature already provides evidence that improved RLHF enhances LLMs in downstream tasks [2] [3]. Our work builds on and compares to existing work [4] [5] in ICLR and similar venues that focuses on preference modeling alone, without downstream LLM finetuning. This facilitates the comparison of our new estimator with SOTA methods in this context. The revised paper significantly expands our numerical study and includes benchmarks from the recent literature (SHP dataset), showing the advantages of our methods; see Table 1 in the revised paper.
>
> We fully concur that downstream LLM evaluation is interesting for future work. A rigorous assessment requires developing a substantial benchmark (as the results may depend on the language model size and other post-training parameters). This is a direction we actively pursue in future research.
>
> > My recommendation would be to either make this a pure theory paper and submit it AISTATS (or a similar venue) or evince the practical benefit of the sign estimator by showing that it works on real human preferences on LLMs of non-trivial size.
>
> Thank you. We applied our sign estimator on two real-world datasets (SHP and Harmfulness/Helpfulness), as you suggested. The performance gaps are significant. We hope that the above discussion clarifies our original set-up.
> While there are several important papers on how to cope with heterogeneity in preference modeling for LLM alignment, they require changes to existing training pipelines (e.g. EM algorithm). The simplicity of the sign estimator—only changing the loss function—is in our view its main practical appeal. Hence, we view ICLR as a good venue to disseminate this idea..

---

> > ### Author Response · Authors · 2025-11-22
> > **Response to Reviewer rvPb [Part 2/2]**
> >
> > [1] Y. Yacoby, W. Pan, and F. Doshi-Velez. Failure modes of variational autoencoders and their effects on downstream tasks. In ICML Workshop on Uncertainty and Robustness in Deep Learning (UDL), 2020.
> >
> > [2] Lingfeng Shen, Sihao Chen, Linfeng Song, Lifeng Jin, Baolin Peng, Haitao Mi, Daniel Khashabi, and Dong Yu. 2023. The trickle-down impact of reward (in-) consistency on rlhf. arXiv preprint arXiv:2309.16155.
> >
> > [3] Leo Gao, John Schulman, and Jacob Hilton. Scaling laws for reward model overoptimization. In International Conference on Machine Learning, pages 10835–10866. PMLR, 2023.
> >
> > [4] Sriyash Poddar, Yanming Wan, Hamish Ivison, Abhishek Gupta, and Natasha Jaques. 2024. Personalizing reinforcement learning from human feedback with variational preference learning. Preprint, arXiv:2408.10075.
> >
> > [5] Anand Siththaranjan, Cassidy Laidlaw, and Dylan Hadfield-Menell. Distributional preference learning: Understanding and accounting for hidden context in RLHF. arXiv preprint arXiv:2312.08358, 2023
> >
> > [6] Bai, Y., Jones, A., Ndousse, K., Askell, A., Chen, A., DasSarma, N., Drain, D., Fort, S., Ganguli, D., Henighan, T. et al. (2022). Training a helpful and harmless assistant with reinforcement learning from human feedback. arXiv preprint arXiv:2204.05862.

---

> > ### Comment · Reviewer_rvPb · 2025-11-25
> >
> > Thanks for looking into my concerns. I appreciate you running the extra experiments. However, the magnitude of improvement in the accuracy is small, even if it may be statistically significant: for most of the experiments with real human preferences, it's <2 percentage points, far less dramatic than the theory and the original experiments would suggest. This only adds to my concerns that the original "digital twins" are not a very faithful reflection of in vivo disagreement.
> >
> > I'm on the fence about this paper, so I will defer to the AC and keep my score as-is for now.

---

> > > ### Author Response · Authors · 2025-11-27
> > >
> > > Thank you for the thoughtful comment. We would like to address 2 points regarding the perceived marginal gain in accuracy:
> > >
> > > > the original "digital twins" are not a very faithful reflection of in vivo disagreement.
> > >
> > > The optimal achievable accuracy is inherently bellow 100%, even with infinite data, due to both idiosyncratic and systematic variability in human preferences. For example, if a dataset is labeled by two individuals with completely opposite preferences, the best possible accuracy is 50%. More generally, the accuracy will be upper bounded by $\mathbb E_{X_1,X_2}[\max(\mathbb P(Y = 1 | X_1, X_2), 1 - \mathbb P(Y = 1 | X_1, X_2))]$, where $\mathbb P(Y = 1 | X_1, X_2)$ is the probability of a *random* individual from the population preferring output 1 over 2. The more diverse the population, the closer is the optimal accuracy to 50%. Characterizing the structure of this variability has been a major focus in research on preference modeling recently [1] [2] [3]. Persona-based simulation has recently emerged as an effective approach for capturing population heterogeneity while producing probabilistic preferences. In Table 1, we observe that **the label accuracy using simulated personas (row 2) is very close to human-labeled data (row 3)**. The primary benefit of persona simulation is that it enables direct comparison to a noise-free ground-truth reward (row 1), allowing clearer interpretation of performance without preference noise.
> > >
> > > > for most of the experiments with real human preferences, it's <2 percentage points
> > >
> > > Predicting noisy human choices is generally difficult, especially when fitting a single reward function to a heterogeneous population. EM, the de facto method for modeling latent heterogeneity, typically yields less than a 1% accuracy improvement in practice, yet such gains are already considered meaningful in the modeling literature. For context, **Google Deepmind's SOTA preference model [4] also reports exactly a 2% increase in accuracy over the RLHF baseline, the same magnitude of improvement discussed here** (please see [4], section G.1, page 23, figure 2). Because label accuracy is inherently limited by human disagreement, small changes in noisy label accuracy can obscure more substantive improvements. This is why examining agreement and other discrepancies with respect to a ground-truth utility, when available, provides a more interpretable view of model gains, as emphasized in our work and others [5, 6].
> > >
> > > We appreciate your valuable feedback and hope that this clarifies the points of concern. Please let us know if you have any remaining questions.
> > >
> > > [1] Joon Sung Park, Carolyn Q Zou, Aaron Shaw, Benjamin Mako Hill, Carrie Cai,
> > > Meredith Ringel Morris, Robb Willer, Percy Liang, and Michael S Bernstein.
> > > Generative agent simulations of 1,000 people. arXiv preprint arXiv:2411.10109,
> > > 2024.
> > >
> > > [2] Shibani Santurkar, Esin Durmus, Faisal Ladhak, Cinoo Lee, Percy Liang, and Tatsunori Hashimoto. Whose opinions do language models reflect? In Proceedings of the 40th International Conference on Machine Learning, pages 29971–30004. PMLR, 2023.
> > >
> > > [3] Olivier Toubia, George Z. Gui, Tianyi Peng, Daniel J. Merlau, Ang Li, and
> > > Haozhe Chen. Database report: Twin-2k-500: A data set for building digital twins
> > > of over 2,000 people based on their answers to over 500 questions. Marketing
> > > Science, 0(0), 2025.
> > >
> > > [4] Remi Munos, Michal Valko, Daniele Calandriello, Mohammad Gheshlaghi Azar, Mark Rowland, Zhaohan Daniel Guo, Yunhao Tang, Matthieu Geist, Thomas Mesnard, Cˆome Fiegel, et al. Nash learning from human feedback. In International Conference on Machine Learning, pages 36743–36768. PMLR, 2024
> > >
> > > [5] Souradip Chakraborty, Jiahao Qiu, Hui Yuan, Alec Koppel, Furong Huang, Dinesh Manocha,
> > > Amrit Singh Bedi, and Mengdi Wang. Maxmin-RLHF: Alignment with diverse human preferences.
> > > arXiv preprint arXiv:2402.08925, 2024
> > >
> > > [6] Paul Gölz, Nika Haghtalab, and Kunhe Yang. Distortion of ai alignment: Does preference optimization optimize for preferences? arXiv preprint arXiv:2505.23749, 2025.

---

### Official Review · Reviewer_D8uD · 2025-11-03

**Soundness:** 3
**Presentation:** 3
**Contribution:** 2
**Rating:** 4
**Confidence:** 4

**Summary:**

This paper addresses the problem of learning an aggregate reward model for LLM alignment from a population of users with heterogeneous preferences. The authors first demonstrate that the standard approach in RLHF, which fits a single probabilistic model to pairwise preference data using cross-entropy loss, yields a biased and inconsistent estimate of the population-average utility. They show that the standard estimator systematically down-weights users with strong, confident preferences. To remedy this, the authors propose a novel method, the sign estimator, which replaces the cross-entropy loss with a binary classification (0-1) loss. This simple modification shifts the learning objective from predicting choice probabilities to merely predicting the correct preference direction (the sign of the utility difference). The contributions come from both theoretical and empirical. Theoretically, the authors prove that under a mild symmetry assumption on the distribution of user preferences, the sign estimator is a provably consistent estimator for the population-average ordinal preferences. They establish the first polynomial-rate finite-sample error bounds for this problem, showing a better convergence rate O(n^{1/3}), a significant improvement over existing rates for general mixture models. Empirically, using a realistic simulation with "digital twins" derived from real user data, the sign estimator is shown to substantially reduce estimation error compared to the standard RLHF baseline, cutting angular error by nearly 35% and disagreement rates with the true population preferences from 12% to 8%. The proposed method also outperforms more complex panel data heuristics that explicitly model heterogeneity, all while maintaining the simplicity of a drop-in replacement within existing alignment pipelines.

**Strengths:**

- [S1] This paper is well-written.

- [S2] RLHF has recently become an active and important research field. Improving the preference modeling can be beneficial for LLM alignment.

- [S3] The theoretical results and simulated experiments support the proposal well.

**Weaknesses:**

- [W1] It is not validated if (1) the sign estimator works in the text preference data in LLMs and (2) can be stable in LLMs finetuning, and (3) the learned preference models benefit RLHF training in LLMs. Also, (4) the title may overstate the contribution ("LLM Alignment" should not be there).

- [W2] Cross-entropy training is actually employed to train not only preference classifiers but also scalar reward models (through Bradley-Terry models), and in practice, the reward models play a more important role in RLHF. In sign estimator, how can we recover reward models?

- [W3] The practical implementation for LLMs is unclear from the paper. Actually some theoretical analysis are based on the assumption of linear model (Assumption 2), which is very different from LLMs.

- [W4] Can't we use  "accuracy" of preference labels as one of the evaluation metrics? It is a bit unclear if better "Angle" and "Disentangle rate" lead to better alignment performance in LLMs.

- [W5] could you explain how EM clustering cluster the preference data more? If my understanding is collect, this paper assumes the preference labels are heterogeneous. So I'm not sure, in that case, how the data is clustered.

- [W6] It is not clear if the real preference data used in LLMs satisfies the assumption of  heterogeneity.

**Questions:**

Please see **Weaknesses** sections above.

---

> ### Author Response · Authors · 2025-11-22
> **Response to Reviewer D8uD [Part 1/2]**
>
> Thank you for the positive feedback and constructive comments. We were pleased to see that you found the paper well written and that you view improved preference modelling as beneficial for LLM alignment.
> > [W1] It is not validated if (1) the sign estimator works in the text preference data in LLMs and (2) can be stable in LLMs finetuning, and (3) the learned preference models benefit RLHF training in LLMs. Also, (4) the title may overstate the contribution ("LLM Alignment" should not be there).
>
> In the revision, we have significantly expanded our numerical study to include standard benchmarks in the literature.
> Evaluating the benefit of improved preference models on downstream LLM alignment (by evaluating a post-trained LLM) is certainly an important performance criterion. However, we build on existing literature that provides evidence that improved preference modelling enhances LLMs in downstream tasks [1] [2] as you also noted. Thus, we focus on this task, as it is in itself an interesting problem. Our empirical set-up enables a comparison with SOTA methods for preference modeling, we follow and compare to prior work, which also focuses on preference modeling from diverse datasets alone, without including experiments for LLM fine-tuning.. This helps evaluate the accuracy gains from using our method, being robust to heterogeneity.
> We fully concur that downstream LLM evaluation is interesting for future work. A rigorous assessment requires developing a substantial benchmark (as the results may depend on the language model size and other post-training parameters). This is a direction we will actively pursue in future research.
> That said, we take note of your point (4) on better qualifying our contribution. We add in the abstract and early on in the paper that our empirical evaluation focuses on preference model performance, rather than LLM performance. We hope this addresses your concern.
> > [W2] Cross-entropy training is actually employed to train not only preference classifiers but also scalar reward models (through Bradley-Terry models), and in practice, the reward models play a more important role in RLHF. In sign estimator, how can we recover reward models?
>
> To clarify, our approach does train and recover a scalar reward model $x\mapsto \Phi(x)^T \hat{\beta}$ or just $x\mapsto R(x)$ in general  (the theoretical recovery guarantee is up to rescaling the reward by a constant, which is absorbed in practice by the KL-penalty temperature during policy optimization). The only difference of our approach is that we replace cross-entropy with a different (Sign) loss function.  We have added a brief clarification in the paper.
>
> > [W3] The practical implementation for LLMs is unclear from the paper. Actually some theoretical analysis are based on the assumption of linear model (Assumption 2), which is very different from LLMs.
>
> We do not need a linear assumption on the LLM. Our approach is agnostic to the LLM. Our consistency results hold for general reward functions R(x) (Theorem 1). Only our strongest theory guarantees (finite-sample complexity) apply to linearly-parametrized reward models. Note, however, that this linear parametrization assumption is not atypical in RLHF, where we append a linear head to a pretrained language model’s hidden representation.
>
> > [W4] Can't we use "accuracy" of preference labels as one of the evaluation metrics? It is a bit unclear if better "Angle" and "Disentangle rate" lead to better alignment performance in LLMs.
>
> We apologize for the confusion. Disagreement (and not Disentangle) rate is simply equal to 1 - Accuracy.
>
> > [W5] could you explain how EM clustering cluster the preference data more? If my understanding is collect, this paper assumes the preference labels are heterogeneous. So I'm not sure, in that case, how the data is clustered.
>
> The EM algorithm iterates between two steps. In the E-step, given a distribution $p$ over $K$ utility vectors $\beta_1, \ldots, \beta_K$, we apply Bayes rule to assign each data point to one of these $K$ clusters probabilistically. Specifically, we compute the posterior $w_{i,k} = \Pr[\beta = \beta_k \mid X_i, y_i, p]$. In the M-step, we update each utility vector using the expected log-likelihood:
> \begin{equation}
> f_k(\beta) = \sum_i w_{i,k} \left[y_i \log(\sigma(\beta^T X_i)) + (1-y_i) \log(1-\sigma(\beta^T X_i))\right].
> \end{equation}
> We then set $\beta_k \leftarrow \arg\max_\beta f_k(\beta)$ and $p_k \leftarrow \left(\sum_i w_{i,k}\right) / \left(\sum_i \sum_k w_{i,k}\right)$. So on and so forth until convergence.
> It is important to clarify that clustering here does not refer to traditional “unsupervised clustering” on covariates. Rather, we assume $K$ latent preference classes and iteratively apply Bayes rule to infer the posterior distribution over these classes for each data point.
> EM is a classical approach for estimating mixture models and has been extensively applied in the RLHF literature to model heterogeneity [3] [4] [5].

---

> ### Author Response · Authors · 2025-11-22
> **Response to Reviewer D8uD [Part 2/2]**
>
> > [W6] It is not clear if the real preference data used in LLMs satisfies the assumption of heterogeneity.
>
> Thank you for this comment; this is an important one.
> Heterogeneity in user preferences is highly significant. LLMs serve a broad user base with diverse preferences, and firms collect preference data to align their models with this population. One might expect that heterogeneity only arises for subjective opinions. But even in the consensual Helpfulness-Harmlessness dataset with trained annotators, [6] reports only 63% agreement between annotators and Anthropic researchers! DONE This level of disagreement, even among annotators who may not fully reflect the broader user population, underscores that preference data are indeed diverse.
>
> Our paper provides additional evidence on the importance of heterogeneity:
>
> Our simulated preference data for a representative panel of 200 digital personas (calibrated to real-world individuals) also comprises significant levels of heterogeneity on the Helpfulness-Harmlessness dataset .
> An indirect measure of heterogeneity is as follows: we apply our estimator (which accounts for heterogeneity) as well as EM on the training dataset and compare their out-of-sample accuracy to that of the standard RLHF method (which does not account for heterogeneity). This new experiment is now included in the revised paper (see Table 1). Both estimators outperform standard RLHF, which would not occur unless there is heterogeneity.
>
> [1] Lingfeng Shen, Sihao Chen, Linfeng Song, Lifeng Jin, Baolin Peng, Haitao Mi, Daniel Khashabi, and Dong Yu. 2023. The trickle-down impact of reward (in-) consistency on rlhf. arXiv preprint arXiv:2309.16155.
>
> [2] Leo Gao, John Schulman, and Jacob Hilton. Scaling laws for reward model overoptimization. In International Conference on Machine Learning, pages 10835–10866. PMLR, 2023.
>
> [3] Sriyash Poddar, Yanming Wan, Hamish Ivison, Abhishek Gupta, and Natasha Jaques. 2024. Personalizing reinforcement learning from human feedback with variational preference learning. Preprint, arXiv:2408.10075.
>
> [4] Keertana Chidambaram, Karthik Vinay Seetharaman, and Vasilis Syrgkanis. 2024. Direct preference optimization with unobserved preference heterogeneity. arXiv preprint arXiv:2405.15065.
>
> [5] Souradip Chakraborty, Jiahao Qiu, Hui Yuan, Alec Koppel, Furong Huang, Dinesh Manocha, Amrit Singh Bedi, and Mengdi Wang. 2024. Maxmin-rlhf: Towards equitable alignment of large language models with diverse human preferences. arXiv preprint arXiv:2402.08925.
>
> [6] Bai, Y., Jones, A., Ndousse, K., Askell, A., Chen, A., DasSarma, N., Drain, D., Fort, S., Ganguli, D., Henighan, T. et al. (2022). Training a helpful and harmless assistant with reinforcement learning from human feedback. arXiv preprint arXiv:2204.05862.

---

> ### Comment · Reviewer_D8uD · 2025-11-28
>
> Thank you for your detailed response and additional experiments. Based on the reward modeling experiments with HH-Anthropic/SHP, and clarification of the scope, I've raised my rating from 4 -> 6. Please also consider the following points:
>
> - Could you change the color of text where you made a changes? That makes us easier to track your changes.
> - Please add the training details of experiments in Table 1 (LLM, hyperparams, compute, etc).
> - Please also consider changing your title. You raised several previos works that argued "improved preference modelling enhances LLMs in downstream tasks" ([1] Lingfeng Shen, Sihao Chen, Linfeng Song, Lifeng Jin, Baolin Peng, Haitao Mi, Daniel Khashabi, and Dong Yu. 2023. The trickle-down impact of reward (in-) consistency on rlhf. arXiv preprint arXiv:2309.16155. [2] Leo Gao, John Schulman, and Jacob Hilton. Scaling laws for reward model overoptimization. In International Conference on Machine Learning, pages 10835–10866. PMLR, 2023.). They are not saying "improving LLM alignment" in the title, rather just mentioning "reward modeling" in the title. When seeing "LLM alignment" in the title, the improvement of LLMs would be expected in the results, which is not included in the current paper.
>
> Note: currently I was not able to edit my "Official Review" in some technical reason. Once it is resolved, I'll update the rating.

---

### Author Response · Authors · 2025-11-22
**Response to all reviewers.**

We thank the reviewers for their positive outlook on the paper and thoughtful comments. Below we address them point-by-point. We highlight our changes to the revised paper in red in the supplementary material.Your feedback helped clarify our contributions:

**Why heterogeneity matters?** There is ample evidence that preference data exhibits heterogeneity [1] [2] [3] and that existing LLM alignment methods are not robust to heterogeneity [4] [5]. Our work builds on this literature and focuses on preference modeling, as it is in itself an important problem. Our analysis brings a new insight: ignoring heterogeneity in preference modeling (or equivalently in direct policy optimization) biases reward models toward preferences of those annotators who care less (or are less informed), this behavior seems undesirable and may have conceptual parallels with observed failures such as sycophancy.

**Empirical setup**: We have significantly expanded the numerical setup and included benchmarks used in previous literature. As recommended by the reviewers, this allows a direct comparison with SOTA methods. The sign estimator outperforms these methods while being an order of magnitude faster to run. Beyond these standard benchmarks, we believe that our new empirical set-up using digital twins can enhance the evaluation of preference models: it captures realistic levels of heterogeneity while providing a known ground truth for utilitarian alignment, enabling more precise performance measurement. Previous work [4] [7] included at most four heterogeneous user types, whereas our approach features preferences calibrated to 200 real-world US panelists. We acknowledge that the value of this new benchmark may not have been sufficiently clear in the initial submission. We believe this semi-synthetic benchmark is itself a contribution, as it enables rigorous assessment of how RLHF methods handle preference heterogeneity.

**Implementation versus theory**: While our approach is justified by theory, our new algorithm design is grounded in practical pipelines. It just amounts to changing the loss function. No architectural change is needed. This is an important deviation from how recent literature copes with heterogeneity. Practical pathways for implementation have been central to this research, and the sign estimator produces a single reward model, similarly to the standard RLHF practice.

**Theoretical contribution**: Our theoretical guarantees constitute a significant new result. Earlier literature [5] [6] proves that distortion is inevitable for general models with heterogeneity. We prove that a simple, practical method can recover preference alignment for a rich class of preference models—our assumption of symmetry subsumes (Gaussian) mixed logits, one of the most widely used models for heterogeneous preferences. Such assumptions appear to be unavoidable, considering the rich previous literature on the estimation of mixture models.

[1] Bai, Y., Jones, A., Ndousse, K., Askell, A., Chen, A., DasSarma, N., Drain, D., Fort, S., Ganguli, D., Henighan, T. et al. (2022). Training a helpful and harmless assistant with reinforcement learning from human feedback. arXiv preprint arXiv:2204.05862.

[2] Michael JQ Zhang, Zhilin Wang, Jena D. Hwang, Yi Dong, Olivier Delalleau, Yejin Choi, Eunsol Choi, Xiang Ren, and Valentina Pyatkin. 2024. Diverging preferences: When do annotators disagree and do models know? Preprint, arXiv:2410.14632

[3] Russel Dsouza and Venelin Kovatchev. 2025. Sources of disagreement in data for LLM instruction tuning. In Proceedings of Context and Meaning: Navigating Disagreements in NLP Annotation

[4] Anand Siththaranjan, Cassidy Laidlaw, and Dylan Hadfield-Menell. Distributional preference learning: Understanding and accounting for hidden context in RLHF. arXiv preprint arXiv:2312.08358, 2023

[5] Paul Gölz, Nika Haghtalab, and Kunhe Yang. Distortion of ai alignment: Does preference optimization optimize for preferences? arXiv preprint arXiv:2505.23749, 2025

[6] Shirali, A., Nasr-Esfahany, A., Alomar, A., Mirtaheri, P., Abebe, R., and Procaccia, A. Direct alignment with heterogeneous preferences. ArXiv, abs/2502.16320, 2025.

[7] Sriyash Poddar, Yanming Wan, Hamish Ivison, Abhishek Gupta, and Natasha Jaques. 2024. Personalizing reinforcement learning from human feedback with variational preference learning. Preprint, arXiv:2408.10075.

---

### Author Response · Authors · 2025-12-04
**Summary of the rebuttal period prior to its interruption**

We would like to summarize our discussion with the reviewers which was converging towards a productive direction:

**Key contributions**:

(1) We give the first RLHF estimator with polynomial convergence rates in the presence of heterogeneous preferences.

(2) the estimator is both theoretically founded and practical: it is a drop-in replacement in existing pipelines (in stark contrast with EM-based approaches) and on large-scale datasets outperforms RLHF and EM type methods by a significant margin.

**Key content from the reviews**:

Reviewer vJr6 focused on our theoretical results. They asked about hidden dimension dependence in our claim of a polynomial rate, and we clarified there was no such dependence, addressing their main concern.

Reviewers D8uD and rvPb both requested additional experiments on large datasets and reporting on additional metrics. In response, we ran further experiments on the SHP and HH Anthropic datasets, and found robust improvements on real human preference data. We also explained why we believe that our original set-up using digital twins was a more realistic and comprehensive benchmark than those used in previous work. In particular, we scale model sizes from 7B to 20B and the heterogeneous population size from 4 to 200 compared to the literature.

In response, reviewer D8uD increased their score. Reviewer rvPb pointed out that the improvement was small. In response to that latter comment, we showed that the improvement is actually significant, both statistically and practically; see our last comment to reviewer rvPb before the discussion stopped due to the current unprecedented circumstances.

Reviewer wnm8 did not engage with the content of our paper; see our separate note on this.

---

### Meta-Review · Area_Chair_mg4o · 2025-12-24

**Summary:**

This paper addresses learning preferences from pairwise comparisons in the presence of heterogeneous users, a critical challenge for aligning large language models (LLMs) with diverse human values. The method proposed is the sign estimator, a novel method that replaces cross-entropy loss with binary classification loss in the aggregation step. The paper argues that this modification results in a consistent and efficient estimator of population-average utility without explicit modeling of user heterogeneity, which mitigates problems in the previous formalism. The method is benchmarked empirically on a semi-synthetic dataset derived from real user data and is shown to outperform standard RLHF approaches in terms of reducing preference distortion.

Reviewers appreciated the clarity of the motivation and the importance of the problem being tackled. The proposed method was praised for its simplicity and practicality as a drop-in replacement for existing alignment pipelines, as well as for its strong motivation (derived from the finding that MLE under heterogeneity amplifies high-variance preferences) and solid
theoretical guarantees in terms of consistency and finite-sample error bounds. A weak aspect in terms of presentation that was noted in the reviews was the unclear positioning of the work within the broader literature on LLM alignment via preference learning and RLHF. This is also reflected by the patchy related work section that does not fully contextualize the contribution within the existing literature. Detachment from the main RLHF and preference fine-tuning literature is also reflected in the empirical evaluation, which was critiqued for not being completely in line with the motivation as the methods hasn't been tested on real human preference data, but on a semi-synthetic dataset derived from real user data. The rebuttals and revisions expand the empirical setup including additional benchmarks from the literature, however the additional experiments still left some reviewers unconvinced about the advantage of the proposed model and its practical relevance for scenarios involving real human preferences.
The detachment of the current work from the main RLHF and preference fine-tuning literature in terms of framing and empirical evaluation contributes to limiting the perceived significance of the contribution. In particular, the paper misses critical discussions and benchmarking against dominant preference fine-tuning methods like PPO and DPO, and methods derived from that literature that also work on unpaired completions, such as  Kahneman-Tversky Optimization (KTO) and Alignment via Optimal Transport (AOT). Absent downstream metrics on mainstream preference benchmarks such AlpacaEval contribute to confirming the opinion that the work's practical impact remains unproven and its broader significance arguably limited.

The recommendation is to build out the empirical evaluation to include real human preference data, which would strengthen the practical relevance of the findings, or alternatively to deepen the theoretical analysis and more thoroughly related it to mainstream preference post-training pipelines and target theory-focused venues.

**Reviewer Concerns:**

* Addressed in the rebuttal:
  - the rebuttals and revisions expand the empirical setup, partially addressing concerns regarding limited empirical evaluation

* Not addressed in the rebuttal:
  - limitations in how the work is positioned within the broader literature on LLM alignment via preference learning and RLHF only partially addressed
  - despite the additional experiments, some reviewers still maintained their concerns that the empirical evaluation is too far removed from practical applications for LLM alignment via post-training and standard RLHF pipelines
  - critical benchmarking against dominant preference fine-tuning methods like PPO and DPO, and methods derived from them are absent, as are downstream metrics on standard evaluation metrics like MT-Bench or AlpacaEval

**Reviewer Scores:**

| Reviewer | initial score | predicted final score |
|---:|---:|---:|
| D8uD | 4 | 6 (explicitly confirmed by reviewer) |
| rvPb | 4 | 4 |
| wnm8 | 2 | 2 |
| vJr6 | 6 | 6 (explicitly confirmed by reviewer) |

---

### Decision · Program_Chairs · 2026-01-26

Reject